# Structural basis for substrate recognition and inhibition of thioredoxin glutathione reductase from *Schistosoma japonicum*: Implications for antiparasitic development

Songqing Wang[1], Wenbin Hong[2], Shukun Zhong[3,4], Zhijian Liang[1], Tianyichen Xiao[1], Chuchu Zhang[1], Xianshu Liu[3,4], Ziyi Dai[3,4], Yunlong Li[1], Siqi Wu[1], Qixu Cai[5], Caiming Wu[1], Yuxuan Huang[2], Peicheng Hong[1], Haixia Ren[1], Shaowei Li[6,7], Tianwei Lin[1]*, Xueqin Chen[2]*, Shuaiqin Huang [3,4]*

1 State Key Laboratory of Cellular Stress Biology, School of Life Sciences, Xiamen University, Xiamen, Fujian, China, 2 Xiamen Key Laboratory of Clinical Efficacy and Evidence Studies of Traditional Chinese Medicine, The First Affiliated Hospital of Xiamen University, School of Medicine, Xiamen University, Xiamen, Fujian, China, 3 Department of Medical Parasitology, Xiangya School of Basic Medical Sciences, Central South University, Changsha, Hunan, China, 4 Key Laboratory of Hunan Province for Schistosomiasis Immunity and Transmission Control, Yueyang, Hunan, China, 5 State Key Laboratory of Vaccines for Infectious Diseases, School of Public Health, Xiamen University, Xiamen, Fujian, China, 6 National Institute of Diagnostics and Vaccine Development in Infectious Diseases, Collaborative Innovation Center of Biologic Products, National Innovation Platform for Industry-Education Integration in Vaccine Research, Xiamen University, Xiamen, Fujian, China, 7 State Key Laboratory of Vaccines for Infectious Diseases, Xiang An Biomedicine Laboratory, School of Life Sciences, School of Public Health, Xiamen University, Xiamen, Fujian, China

* twlin@xmu.edu.cn (TL); xqchen@xmu.edu.cn (XC); sqhuang@csu.edu.cn (SH)

## Abstract

Praziquantel (PZQ) is currently the only agent for treating schistosomiasis, but it is plagued by suboptimal efficacy to juvenile parasites, looming drug resistance, and inability to prevent reinfection. Thioredoxin glutathione reductase (TGR) is regarded as a promising therapeutic target due to its essential role in maintaining schistosome redox homeostasis. Herein, the crystal structures of *Schistosoma japonicum* TGR (SjTGR) in multiple redox states and in complex with NADPH, GSH, and the anti-helminthic agent Auranofin were elucidated. Structural analyses identified the hook-shaped conformation at the C-terminal redox center, which DTNB assays further confirmed enhances electron transfer efficiency. Structural and ITC data indicated that R317 was critical for NADPH binding via hydrogen-bond interactions. The analysis also indicated that the structure basis of Auranofin's potency was its tripartite interaction at the redox-active sites. In addition, we investigated the substrate specificity of SjTrx1i and SjTRP14, downstream proteins regulated by SjTGR, and elucidated the structural basis for this specificity by determining their oxidized/reduced structures. Furthermore, *in vivo* RNAi indicated knockdown of SjTGR or SjTRP14 blocked the survival and oviposition of schistosomes, thus ameliorating

**Data availability statement:** The atomic coordinates of the structure and the structure factors were deposited in Protein Data Bank with PDB ID codes 22EY, 22FC, 22FD, 22FE, 22FF, 22FG, 22FH,22FJ, 22FK, 9LWM and 9LWZ.

**Funding:** This work was supported by grants from the National Natural Science Foundation of China (nos. 82102428 to S.H.), the Natural Science Foundation of Hunan Province (nos. 2022JJ40663 to S.H.), and the Open Research Project of the Regional Collaboration Centre for Schistosomiasis Control Technology in Lake Regions for the Year 2025-2026 (nos. 20254003 to S.H.). The funders had no role in study design, data collection and analysis, decision to publish, or preparation of the manuscript.

**Competing interests:** The authors have declared that no competing interests exist.

egg-induced granulomatous pathology in mice. This work provided a framework for knowledge-based design of novel anti-schistosomals targeting parasite-specific redox vulnerabilities.

## Author summary

Schistosomiasis, a devastating neglected tropical disease (NTD), affects approximately 240 million individuals and puts over 700 million at risk. While praziquantel (PZQ) is the only available therapeutic, it faces significant limitations, highlighting an urgent need to discover novel drug targets and develop new anti-schistosomal agents. Thioredoxin glutathione reductase (TGR), essential for parasite survival, is a promising target; however, structural insights into *S. japonicum* TGR (SjTGR) remain limited, hampering targeted drug development. Here, we report the complete catalytic cycle of SjTGR, resolved by X-ray crystallography, establishing the structural basis of its mechanism. Functional studies confirm its critical role in maintaining schistosome survival and fecundity. By elucidating its unique catalytic architecture including a tripartite active site, T594-mediated electron transfer enhancement, and R317-dependent NADPH stabilization, this work reveals schistosome-specific redox adaptations. These structural insights distinguish SjTGR from its mammalian orthologs and provide a robust foundation for designing selective therapeutics against schistosomiasis, therefore contributing to the global elimination of this disease burden.

## 1 Introduction

Schistosomiasis, caused by parasitic flatworms of the genus *Schistosoma*, is a devastating but neglected tropical disease [1,2]. There are about 240 million people affected in no less than 78 countries and over 700 million people at risk of infection [3,4]. Currently, the primary strategy to control schistosomiasis is by periodic and large-scale preventive chemotherapy of at-risk populations with praziquantel (PZQ) which is currently the only available drug to treat schistosomiasis [4,5]. However, several critical problems persist, including low cure rates, limited efficacy against juvenile liver-stage worms, ineffective prevention of reinfection, difficulty for treating school-aged children, not to mention the looming PZQ resistance [6–10]. It is of great urgency to identify novel drug targets and develop new anti-schistosomal agents for treating the infected and control the endemic.

In most eukaryotic organisms, the glutathione (GSH) and the thioredoxin (Trx) systems are two primary and independent pathways responsible for the thiol-dependent redox processes including maintaining the cellular redox homeostasis, antioxidant defenses, protein folding, and electron transfer for the deoxyribonucleotide synthesis [11–13]. As the key antioxidant enzymes, glutathione reductase (GR) transfers electrons from NADPH to the oxidized tripeptide glutathione (GSSG), whereas

thioredoxin reductase 1 (TrxR1) transfers electrons from NADPH to both oxidized thioredoxin 1 (Trx1) and the thioredoxin-related protein of 14 kDa (TRP14) [12]. TrxR1 undergoes sequential reduction by two equivalents of NADPH, proceeding from the fully oxidized state ($E_{ox}$) to the two-electron-reduced intermediate ($EH_2$) with reduced $FADH_2$, and finally to the four-electron-reduced enzyme ($EH_4$), thereby enabling substrate reduction [12]. However, unlike Trx1, TRP14 cannot reduce known Trx1 substrates, including ribonucleotide reductase, peroxiredoxin, or methionine sulfoxide reductase [12]. Between these two systems, TrxR is considered to be an important drug target for various ailments, including infectious diseases and cancers [11–12]. In contrast to their mammalian hosts and free-living platyhelminths, the GSH and Trx systems in schistosomes, represented by *S. mansoni* and *S. japonicum*, are intertwined and, instead of depending on two separate enzymes, the two GR and TrxR activities are conjoined into a single enzyme, a selenocysteine (Sec/U)-containing thioredoxin glutathione reductase (TGR), for maintaining the thiol-disulfide homeostasis [13–15].

TGR is indispensable for the survival of the parasite and a valid and promising drug target for schistosomiasis [16–19]. To control the disease, novel compounds targeting TGR have been developed, including Auranofin, oxadiazole 2-oxides and its derivatives, Furoxan, 1,4-naphthoquinone ether derivatives, and several non-covalent inhibitors [10,20–23]. These compounds not only provide drug candidates to treat the disease, but also offer tools for in-depth studies of TGR's reaction mechanism [24].

*S. mansoni* TGR (SmTGR) is a Sec-containing and homodimeric flavoenzyme. In conjunction with its dual activity, SmTGR is comprised of two homo-subunits and each subunit is a structural conjugate of a glutaredoxin (Grx) domain and a TrxR [25]. The Sec residue essential for the catalytic activity is in a GCUG tetrapeptide at the C-terminus (S1 Table) [25,26]. The catalytic cycle of TGR is of a reductive half-reaction and an oxidative half-reaction [10,26]. After binding to TGR, NADPH donates electrons from its nicotinamide ring to FAD interacting with TGR with its isoalloxazine ring and the nearby C154/C159 redox couple [10,26] for transferring the electrons first to the C596/U597 redox couple on the C-terminal segment of the neighboring subunit, and then to the oxidized Trx or the C28/C31 redox couple of the neighboring monomer to reduce GSSG (S1 Table) [10,26,27]. Another TGR under investigation is *S. japonicum* TGR (SjTGR) (PDB: 4LA1) [28–30], which though was with the structurally undefined C-terminal redox center (S1 Table). As only the apo structure is available for SjTGR, it was of great interest to determine the complex structures of SjTGR with substrates and inhibitors to understand their catalytic mechanism and establish a structural framework for developing therapeutics to control the endemic and treat the disease.

In this study, the biochemical and structural properties of SjTGR complexes with its native substrates, and cofactors NADPH and GSH, as well as an inhibitor, Auranofin, were investigated, which provided snap shots in various stages of the catalytic cycle. A distinct hook-shaped conformation in the C-terminal redox center of SjTGR was revealed with the key residue R317 binding NADPH. The structural analysis indicated that Auranofin inhibited SjTGR's enzymatic activity by a tripartite mechanism. The complex structures of SjTGR with its physiological substrates SjTrx1i and SjTRP14 provided insights into their distinct substrate recognition and binding patterns. By incorporating data on enzymatic kinetics and functional studies, this work provided the biochemical and structural basis for the substrate and inhibitor recognition by SjTGR and shed light on the schistosome-specific redox adaptations and established a structural framework for developing therapeutics targeting SjTGR.

## 2 Results

### 2.1 Structural basis of the electron transfer in SjTGR

Similar to SmTGR, SjTGR is a homodimer and each monomer is equivalent to fusion of a Grx domain (residues 1–105) and a TrxR (residues 106–598 in SjTGR). TrxR is comprised of a linearly discontinuous FAD binding domain (residues 106–286, 394–463), an NADPH binding domain (residues 287–393), and a dimerization domain (residues 464–598) (Figs 1A, S1A; S1 File). The crystal structures of SjTGR^WT and SjTGR^U597C (a variant with the penultimate Sec mutated to cysteine) were determined (S1B-S1I Fig; S3 Table). The dimeric SjTGR was a reminiscent of the TGR's distorted

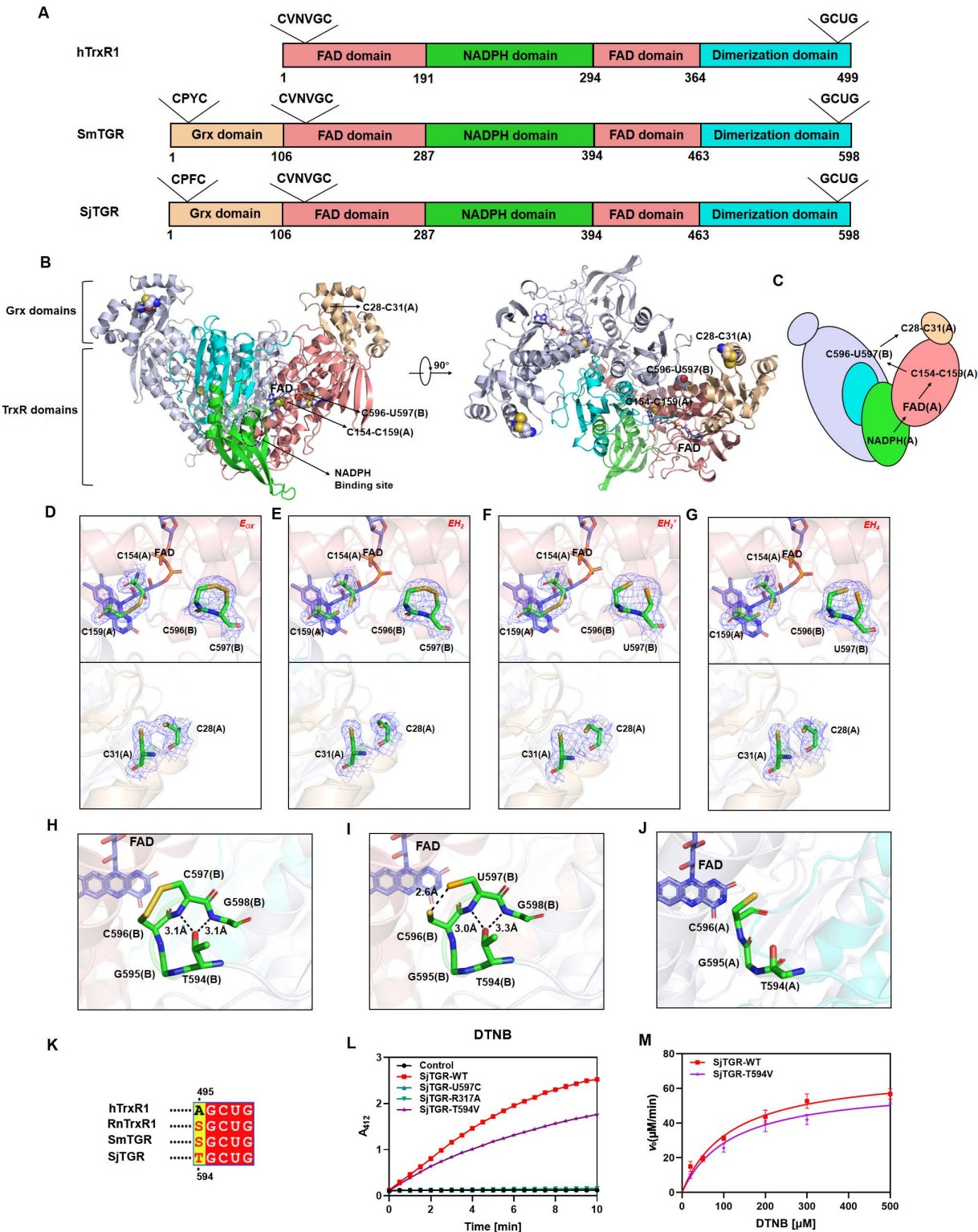

**Fig 1. The entire electron transfer cycle of SjTGR. (A)** Domain architecture comparison of human TrxR1 (hTrxR1), *Schistosoma mansoni* TGR (SmTGR), and SjTGR. Domains were color-coded: Grx (light orange), FAD-binding (salmon), NADPH-binding (green), and dimerization (cyans).

**(B)** The overall structure of SjTGR (PDB: 22EY) was represented in a cartoon format, with domain coloring in subunit A consistent with **(A)**. In subunit B, the Grx domain and TrxR were colored domain light blue. The FAD cofactor was depicted as slate sticks, while three redox-active centers (C154-C159, C596-U597, and C28-C31) were shown as spheres, and the NADPH binding site highlighted with a dashed circle. **(C)** Electron transport model of SjTGR. **(D-G)** The $2Fo\text{-}Fc$ electron density maps (blue mesh, contoured at $1\sigma$) for the three redox centers of SjTGRWT (GCUG) and SjTGRU597C (GCCG) are shown. Average B-factors ($\text{Å}^2$) for C28–C31, C154–C159, and C596–C/U597 are provided with their corresponding PDB IDs: (D) 9LWZ (56.68, 33.54, 55.95), (E) 22FC (55.33, 37.90, 64.09), (F) 22EY (67.70, 38.83, 68.13), and (G) 9LWM (58.89, 33.88, 56.07). **(H-I)** Final five residues of subunit B (green sticks) in oxidized (H) (PDB: 22FC) and reduced (I) (PDB: 22EY) states, and distance measurements are indicated by black dashed lines. **(J)** The green sticks representation of C-terminal residues (T594-C596) in subunit A of SjTGR (PDB: 22FC). **(K)** Sequence alignment of the C-terminal redox motif from structurally characterized TGR and TrxR homologs including *Homo sapiens* TrxR1 (hTrxR1, NCBI accession: NP_877393.1), *Rattus norvegicus* TrxR1 (RnTrxR1, NCBI accession: O89049.5), *Schistosoma mansoni* TGR (SmTGR, NCBI accession: XP_018649018.1), and *Schistosoma japonicum* TGR (SjTGR, NCBI accession: ACH86016.1). **(L)** Enzyme activity assays for SjTGR[WT] and the mutants SjTGR[U597C], SjTGR[T594V] and SjTGR[R317] were performed using DTNB as the substrate, measuring TNB production at 412 nm, with no SjTGR as the control group (mean ± SD, n = 3; In some cases, error bars are smaller than the symbol and thus not visible). **(M)** Enzyme kinetic curves of SjTGR[WT] and SjTGR[T594V] (mean ± SD, n = 3).

W-shaped architecture from parasitic platyhelminths [25]. This W shaped could be considered as comprised of two V with the dimerization domain forming the downward stroke of the V, NADPH binding domain making the turn, and FAD binding domain forming the other stroke with the Grx domain placed atop. The interaction of the dimerization domain brought two V shaped subunits into a W shaped molecule to form the homodimeric SjTGR (Fig 1B). In this structure, NADPH transferred the electrons sequentially to FAD, to the nearby C154-C159, then to the C-terminal selenothiol at the active site of the neighboring subunit, and ultimately to either the substrate for the catalytic reduction in the neighboring subunit or back to the redox-active C28-C31 motif in the Grx domain (Figs 1B- 1C, S2A- S2B). Superposing SjTGR onto SmTGR (PDB: 7B02), there was a 5.7° rotation between the two Grx domains (S2C Fig). This difference between the two enzymes resulted in a distal difference of 3.0 Å between the catalytic cysteine residues (C28) in the Grx domains. It was so happened that the distance between the C-terminal selenocysteine (U597) in the neighboring subunit was 3.2 Å. As a result, the distance for transferring the electrons between C28 and U597 were comparable in the two enzymes, as that distance for SjTGR was 35.8 Å and 34.6 Å for SmTGR (S2D Fig). The conserved spatial arrangement of redox centers in *Schistosoma* species underscored the stringent evolutionary constraints imposed on the inter-subunit electron transfer pathway, which was crucial for maintaining TGR functionality in parasitic flatworms [14,43,44].

The structures of SjTGR[WT] and SjTGR[U597C] in four redox states were analyzed, which included the fully oxidized state ($E_{ox}$), two intermediate states reduced with two electrons ($EH_2$), and the fully reduced state with four electrons ($EH_4$) (Figs 1D-1G, S3A-S3D). Distinct TrxR conformations could be correlated to the series of redox states. From $E_{ox}$ to $EH_4$, the N-terminal redox center (C154-C159) in $E_{ox}$ accepted electrons from FAD in $EH_2$, which were transferred to the C-terminal redox center (C596-U597) of the adjacent subunit in $EH_2$. As the N-terminal redox center (C154-C159) accepted electrons again, the protein was in the fully reduced state of $EH_4$. During these sequential molecular events, the catalytic cysteines (C28-C31) in the Grx domain remained reduced across all states, suggesting that the substrate reduction process mediated by the Grx domain did not completely depend on the redox cycle of TrxR. Intriguingly, the C-terminal selenothiol motif (C596-U597/C597) in SjTGR adopted a cis-conformation in both oxidized and reduced states (Figs 1D- 1G, S3A-S3D), in contrast to the redox-dependent trans-cis transitions in *Rattus norvegicus* TrxR (RnTrxR1) (PDB: 3EAN) and SmTGR (PDB: 7B02) (S3E- S3G Fig). Further analysis indicated that this unique feature of fixed configuration was mediated by T594, which formed a hydrogen bond with the backbone nitrogen of U597 (or C597 in the U597C mutant) and G598, anchoring the selenothiol motif into a distinct hook-shaped conformation (Figs 1H- 1I, S3H-S3I). There was also a conformational variation between the subunits with the electron density defined for residues 597–598 in subunit B, but not for that in subunit A (Figs 1J, S3J). As a threonine residue was required to form this hook-shaped conformation, which was unique to SjTGR as it is alanine for hTrxR1, serine for RnTrxR1/SmTGR (Figs 1K, S1A), this T594-stabilized conformation was unique to SjTGR.

Structural analyses of SjTGR$^{WT}$, SjTGR$^{T594V}$ and SjTGR$^{U597C}$ indicated that the C-terminal redox motif played a critical role in catalysis and electron relay. Further enzymatic investigation showed that the wild-type enzyme exhibited a robust DTNB reductase activity, and SjTGR$^{U597C}$ with a mutation on the critical selenol group had no detectable redox catalytic activity under the same assay conditions (Fig 1L), demonstrating that the selenol group is essential for SjTGR enzymatic function. A T594V mutation that would interfere on the hook-shaped conformation reduced the catalytic efficiency ($K$cat/$K$m) by 13%, from 12.07 µM$^{-1}$·min$^{-1}$ to 10.45 µM$^{-1}$·min$^{-1}$ (Fig 1L- 1M; Table 1), indicating that T594 contributes to maintaining the optimal geometry for electron transfer.

## 2.2 Conformational dynamics of R317 stabilizes SjTGR-NADPH binding

The structure of SjTGR complexed to NADPH was determined to a resolution of 1.90 Å (PDB: 22FD). There was one NADPH molecule in each subunit. NADPH bound at in a pocket of high hydrophilicity in the NADPH domain, at the interface with the FAD domain (Fig 2A, S4 Table). In contrast to the FAD binding, the NADPH-binding pocket was in a surface-exposed cleft (Fig 2B). Only the electron density for the 2'-monophosphoadenosine-5'-diphosphate portion of NADPH was defined, indicating that NADPH was not tightly bound and could be readily dissociable from the enzyme (Figs 2C, S4A). Comparing to the fully defined NADPH density in SmTGR (PDB: 2X99), *Brugia malayi* TrxR1 (BmTrxR1) (PDB: 7PUT), and RnTrxR1 (PDB: 3EAN), this NADPH and SjTGR complex structure could be an intermediate with the NADPH molecule entering or exiting the binding pocket. In the complex structure between NADPH and SmTGR, the phenolic ring of Y296 was paralleled to the isoalloxazine ring of FAD bound nearby, while the phenolic ring of Y296 of SjTGR was perpendicularly to the isoalloxazine ring of the FAD molecule (Figs 2C-2D, S4). This was an indication that the binding of NADPH involved a conformation change in Y296. Comparing the NADPH-bound and apo SjTGR structure, R317 also adopted two distinct conformations in apo SjTRG and NAPDP-bound SjTRR to stabilize the adenine ring (Figs 2C-2E, S4). The SjTGR binding of NADPH also resulted in the formation of hydrogen bonds with residues S318 and R322 (Fig 2F).

As R317 seemed to be critical for SjTGR binding of NADPH, the residue was mutated for ITC and enzymatic activity assays. The binding of NADPH to SjTGR was characterized as an endothermic reaction at 25°C. The SjTGR$^{WT}$ bound NADPH with an affinity $Kd = 1220 ± 492$ nM and could mediate the reduction of DTNB. In contrast, SjTGR$^{R317A}$ lost the ability to bind NADPH and failed to reduce its substrate (Figs 2G-2H, S5, 1L). So R317 was essential for the SjTGR binding of NADPH and its enzymatic activity.

## 2.3 The reduction of GSSG and GSH-dependent reductive activity

The three-dimensional structure of the SjTGR/GSH complex was determined to a resolution of 2.12 Å (PDB: 22FE) (S4 Table). In contrast to the electron density with NADPH, only one of the Grx domains (subunit A) of SjTGR was with a bound GSH molecule (Fig 3A). The N-terminal glutamate of GSH formed a hydrogen bond with S85 of SjTGR at a distance of 2.2 Å, and the C-terminal glycine of GSH is stabilized by hydrogen bonds with K25 and Q60 of SjTGR at distances of 3.1 Å and 2.9 Å, respectively (Fig 3B). The formation of a hydrogen bond between residue 72 and the sulfur

**Table 1. Kinetic parameters of SjTGR-WT and SjTGR-T594V with DTNB.**

| Enzyme | SjTGR$^{WT}$ | SjTGR$^{T594V}$ |
|---|---|---|
| Substrate | DTNB | DTNB |
| $K$m (µM) | 116.90 (95% CI: 85.88–159.50) | 118.80 (95% CI:84.81–167.50) |
| $V$max (µM·min$^{-1}$) | 70.55 (95% CI: 63.56–79.17) | 62.09 (95% CI: 55.36–70.63) |
| $K$cat (min$^{-1}$) | 1411.00 (95% CI: 1271.20–1583.40) | 1241.80 (95% CI: 1107.20–1412.60) |
| $K$cat/$K$m (µM$^{-1}$ min$^{-1}$) | 12.07 | 10.45 (13.42% ↓) |

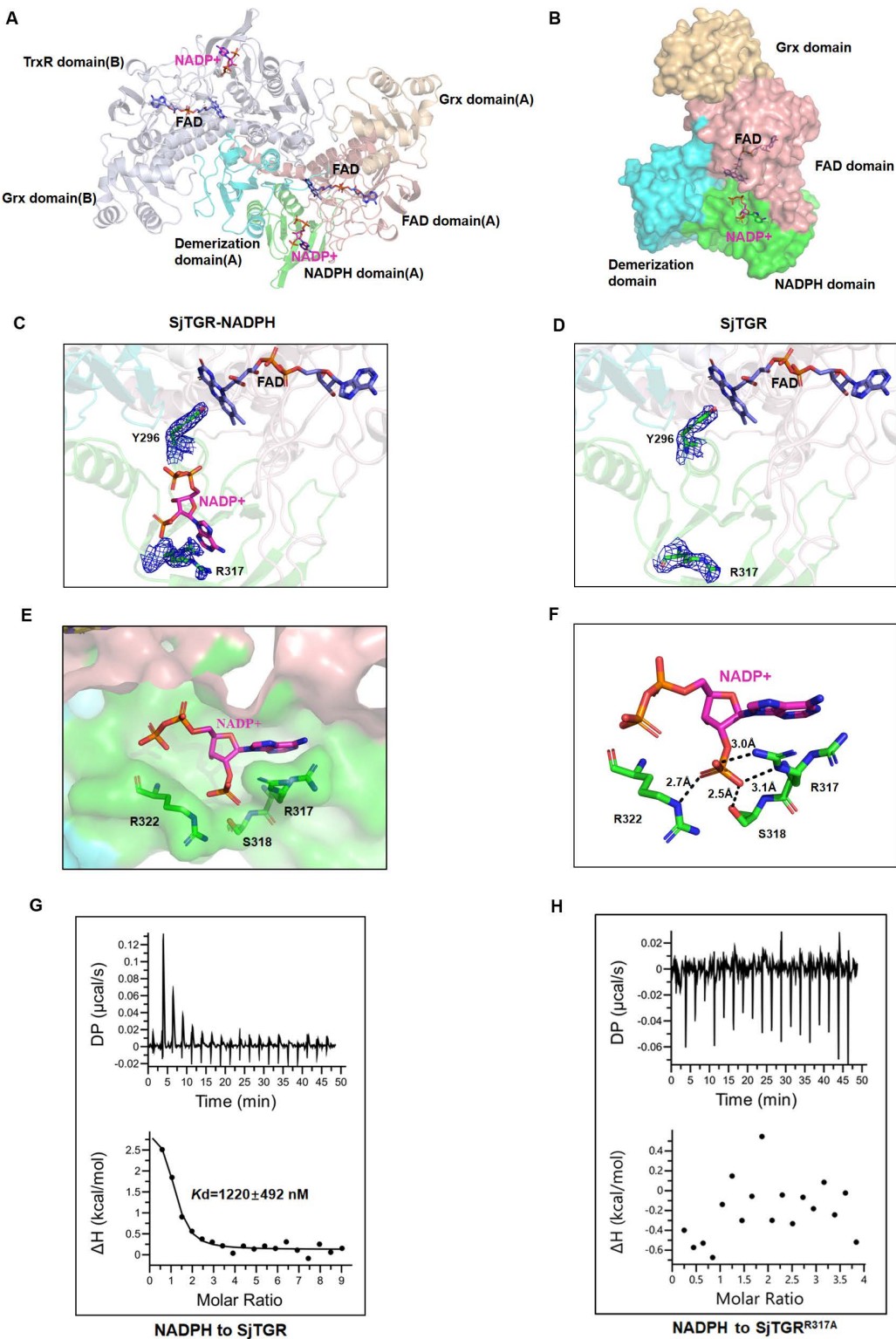

**Fig 2. Structural and thermodynamic basis of NADPH binding in SjTGR.** (A) Cartoon representation of the SjTGR (PDB: 22FD) homodimer bound to NADPH (violet sticks). (B) Surface diagram of SjTGR (PDB: 22FD) binding with NADPH, highlighting domain organization. (C-D) The 2*Fo-Fc* electron

density maps for Y296 and R317 in SjTGR are shown, contoured at 1σ in blue meshes. Panel (C) depicts the NADPH-bound state of SjTGR (PDB: 22FD), while panel (D) shows the unbound state (PDB: 22EY). (E) The surface diagram shows the spatial immobilization effect of R317, S318 and R322 on NADPH (PDB: 22FD). (F) Detailed interactions between NADPH and residues R317, S318, and R322 in SjTGR (PDB: 22FD) are depicted, with these residues shown as green sticks. Distance measurements are represented by black dashed lines. (G-H) ITC measurements of NADPH binding to SjTGR and its mutant SjTGR[R317A]. Shown are representative images from triplicate measurements.

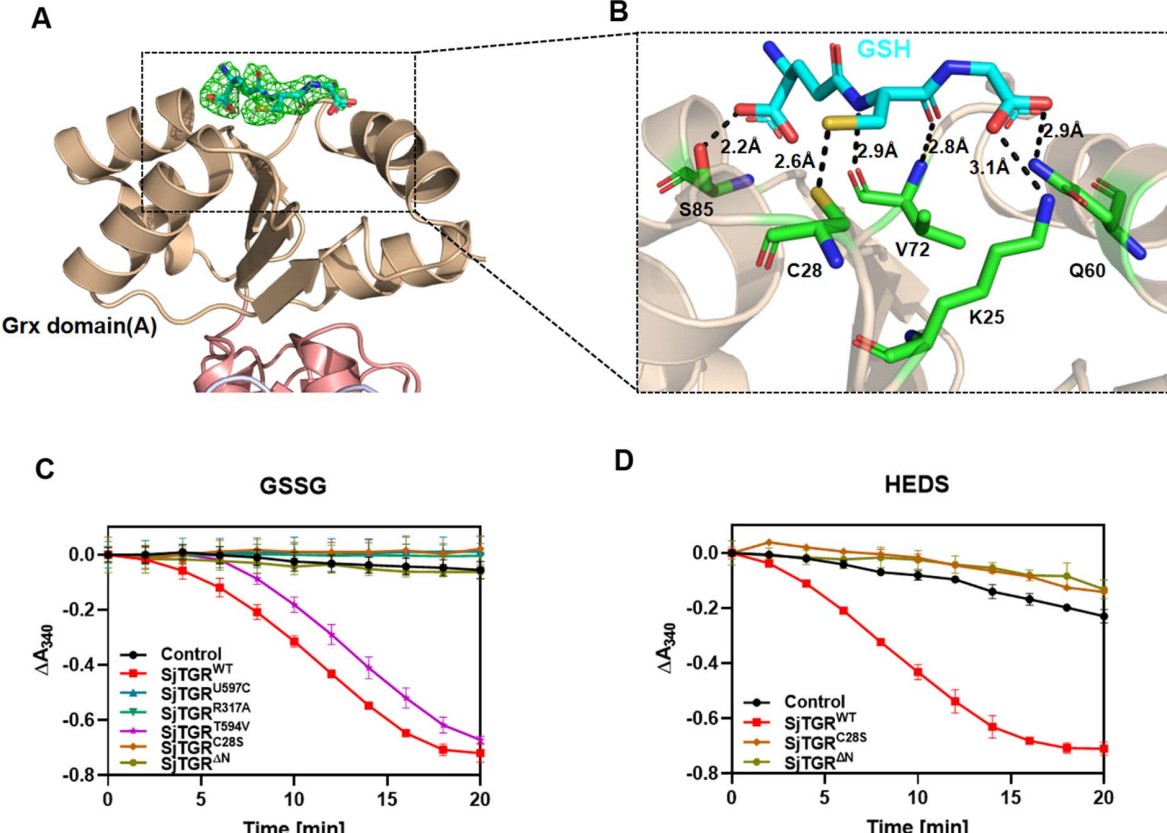

**Fig 3. The Grx domain is capable of reduction by GSH and concurrently reducing GSSG.** (A) Interactions between GSH and Grx domain of SjTGR (PDB: 22FE). The *Fo-Fc* electron density map for GSH is contoured at 3σ and displayed in green meshes. GSH is shown as blue sticks. (B) Residues involved in GSH binding are shown as green sticks. Distance measurements are shown as black dashed lines. (C) Enzyme activity assays were performed for SjTGR[WT], SjTGR[U597C], SjTGR[T594V], SjTGR[C28S], and SjTGR[ΔN]. The reduction of NADPH was measured at 340 nm using GSSG as the substrates, with no SjTGR included as the control group (mean ± SD, n = 3; In some cases, error bars are smaller than the symbol and thus not visible). (D) Reduction curves of HEDS by SjTGR[WT], SjTGR[C28S], and SjTGR[ΔN], with no SjTGR included as the control group (mean ± SD, n = 3; In some cases, error bars are smaller than the symbol and thus not visible).

atom in GSH resulted in the orientation of the sulfur atom toward the redox active site of the Grx domain (2.6 Å distance from C28 sulfur) (Fig 3B), which was responsible for the delivery of reducing equivalents.

To delineate the functional contributions of the Grx domain and its active-site residue C28, two additional mutants, SjTGR[C28S] and SjTGR[ΔN] (a SjTGR mutant with residues 1–105 deleted from the Grx domain), were generated. The GR activity assays using GSSG as the substrate revealed that SjTGR[U597C], SjTGR[R317A], SjTGR[C28S], and SjTGR[ΔN] completely lost the NADPH-dependent GSSG reductase activity, while SjTGR[T594V] retained the activity (Fig 3C). This demonstrated that TrxR alone could not directly transfer electrons from NADPH to GSSG, nor could it bypass the Grx domain to engage

alternative redox relays. The assays with hydroxyethyl disulfide (HEDS) as the substrate corroborated the notion that C28 was indispensable for the activity. SjTGR$^{WT}$ was with robust HEDS reductase activity, but the activity was abolished in SjTGR$^{C28S}$ and SjTGR$^{ΔN}$ (Fig 3D). These results in conjunction with the crystallographic data showed that the reduced GSH initiated the electron transfer to downstream substrates exclusively through C28 in the Grx domain.

Together, these findings demonstrated that the Grx domain, specifically its redox-active C28 residue, served as an obligate electron conduit for both GR and Grx functions in SjTGR. The complete loss of activity in SjTGR$^{ΔN}$ underscored the domain's non-redundant role in coordinating the dual catalytic activities, while the partial retention of the activity in SjTGR$^{T594V}$ was an indication that the residue was involved in conformational stabilization rather than direct participation in the catalytic activity. This mechanistic dichotomy highlighted the evolutionary optimization of SjTGR as a multifunctional redox hub in schistosomes, and positioned the Grx domain as a compelling target for species-specific inhibitor design.

## 2.4 Auranofin inhibits SjTGR through triple redox center targeting

The pan-TrxR inhibitor auranofin (Au) is a potent anti-schistosomal agent and inhibits the SjTGR enzymatic activity as well as the parasite viability *in vitro* and *in vivo* [18]. However, its structure basis of its interactions with TGR remained elusive. The complex structure between SjTGR and Au was determined to a resolution of 1.84 Å (PDB: 22FF) (S4 Table). It was a surprise that Au bound at all three redox centers: (1) The N-terminal Grx domain active site (C28-C31), (2) Inter-subunit electron transfer axis (C154-C159), and (3) C-terminal selenothiol motif (C596-U/C597). Each Au atom bridges adjacent cysteine thiols (2.3-2.5 Å coordination bonds), preventing disulfide bond formation and blocking electron transfer (Fig 4A-4C). This represents the first structural evidence of an Au simultaneously targeting three catalytic centers for a flavoreductase to inhibit both TrxR and Grx functional modules (S5 Table). The Au molecule interacted with C597 with defined density in the subunit B; however, T594 did not form a hydrogen bond with the N atom of the C597 peptide bond (4.1 Å) (Figs 4D, S6A-S6B), consistent with the absence of electron density for the final two amino acids, suggesting subunit asymmetry arising from the hydrogen bond interaction of T594. In contrast, in the subunit B, T594 maintained a hydrogen bond with the N atom of the C597 peptide bond (3.1 Å) (Figs 4E, S6C-S6D), indicating that Au could not completely disrupt this interaction, although the electron density corresponding to the terminal nonconserved amino acid G598 was absent, possibly due to Au-induced disruption of the T594-G598 hydrogen bond.

To assess the effects of different inhibitors on the enzymatic activity of SjTGR, we obtained another thioredoxin reductase inhibitor, TRi-1, as well as a previously reported SjTGR inhibitor, Sj001 [31] (Figs 4F, S7A-S7B). All three inhibitors interfered with the enzymatic activity of SjTGR, but Au was the most potent (Figs 4G, S7C). Sj001 bound SjTGR with a $K$d value of 11.2±3.8 μM (Figs 4H, S8). These findings establish Au's unique multi-target inhibition mechanism, highlighting the therapeutic potential of exploiting SjTGR's non-mammalian structural features, particularly the tripartite redox center architecture, for selective antischistosomal drug design.

## 2.5 Structural bases for substrate specificities of SjTGR's downstream substrates SjTrx1i and SjTRP14

Trx is a typical substrate of TrxR for maintaining its reduced state. Trx-related protein 14 (TRP14), a new member of Trx family, has similar redox potential to that of Trx and receives the electrons from TrxR1 [32,33]. The enzymatic assays were employed to evaluate the potential of SjTrx1i and SjTRP14 as the endogenous substrates for SjTGR. The cDNAs encoding an SjTrx1 isoform (SjTrx1i) and SjTRP14 were cloned from a cDNA library of *S. japonicum* and the recombinant proteins were expressed and purified (S9A-S9H Figs, S1 File). This SjTrx1i is with five amino acid differences to the previously reported SjTrx1 gene (S9I Fig). Using SjTrx1i and SjTRP14 as substrates, the catalytic activity of SjTGR was analyzed by measuring the rate of NADPH reduction. It was demonstrated that SjTGR could reduce both SjTrx1i and SjTRP14, but the catalytic rate for SjTrx1i was significantly higher over that for SjTRP14 (Fig 5A). SjTGR transfers electron to SjTrx1 which then could reduce insulin [34,35]. By quantifying the precipitation of insulin at a wavelength of 650 nm, the reductive capacity of various SjTGR mutants on SjTrx1i could be evaluated. It could be shown that despite exhibiting

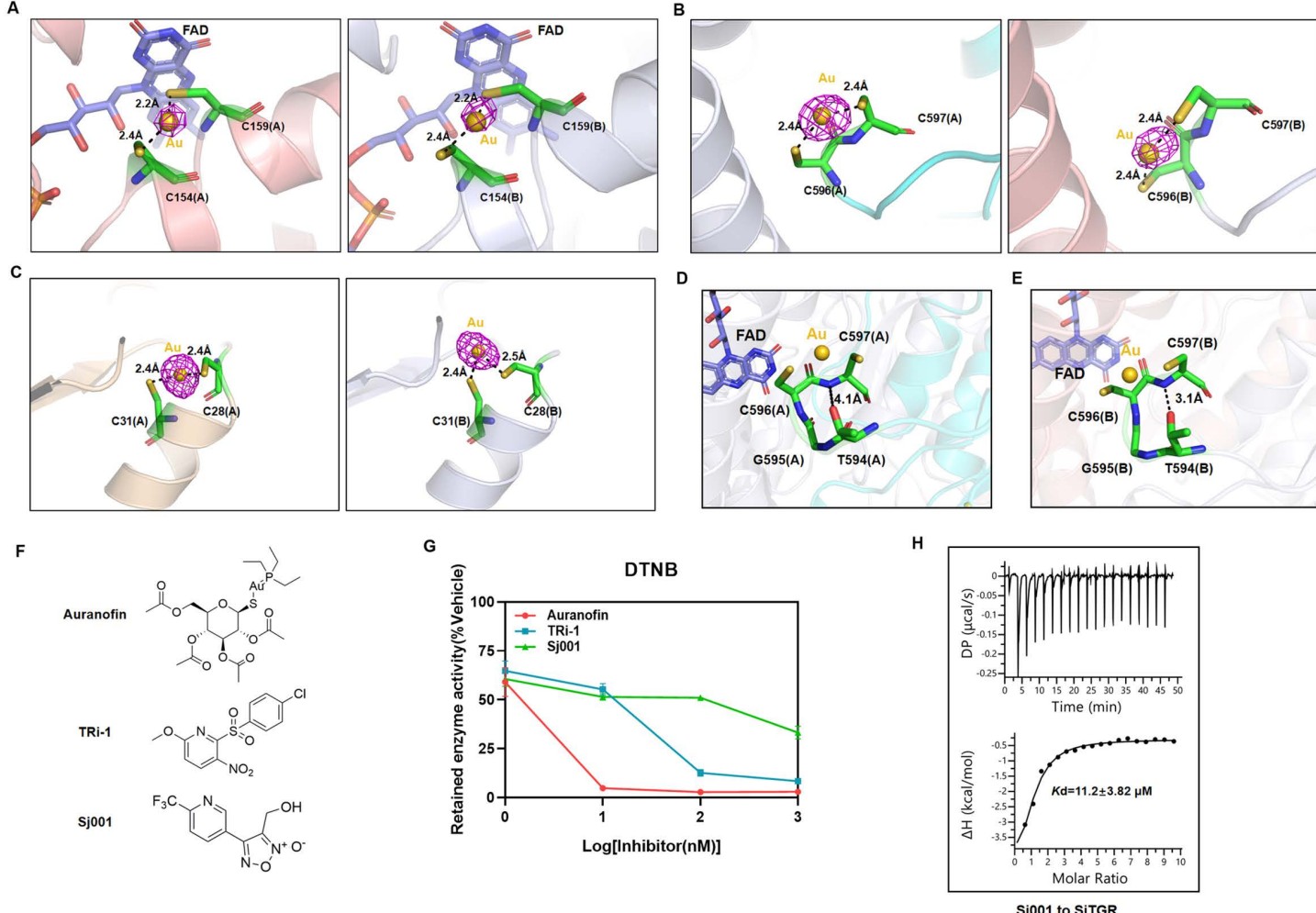

**Fig 4. Inhibitory effects of Auranofin, TRi-1, and Sj001 on SjTGR.** (A-C) Details of the interactions between Au and SjTGR (PDB: 22FF) in three redox centers: C154-159 redox center (A), C596-U/C597 redox center (B) and C28-C31 redox center(C). Au is represented as gold spheres, the residues on SjTGR interacting with Au are shown as green sticks, the distance measurements are depicted as black dashed lines, and the anomalous scattering signal of Au contoured at 7σ is shown as violet meshes. (D-E) The conformations of the C-terminal of SjTGR (PDB: 22FF) subunits A and B (green sticks) are illustrated under the influence of Au, with distance measurements indicated by black dashed lines. (F) Chemical structures of Auranofin, TRi-1, and Sj001. (G) Inhibitory effects of different concentrations of Auranofin, TRi-1, and Sj001 on SjTGR enzyme activity, using DTNB as the substrate (mean±SD, n=3; In some cases, error bars are smaller than the symbol and thus not visible). (H) ITC analysis of Sj001 binding to SjTGR at 25°C. Shown are representative images from triplicate measurements.

catalytic efficiencies similar to SjTGR[T594V], SjTGR[ΔN] and SjTGR[C28S] could still reduce SjTrx1i (Fig 5B), suggesting that neither the Grx domain nor the C28-C31 motif was necessary for the SjTGR reduction of SjTrx1i. In contrast, SjTGR[U597C] and SjTGR[R317A] failed to reduce SjTrx1i (Fig 5B), indicating that the Sec residue and NADPH were essential for SjTGR to transfer electrons to SjTrx1i.

Although hTrx1 can reduce insulin, hTRP14 cannot, despite both can reduce both oxytocin and vasopressin [32]. The insulin reduction by either SjTrx1i and SjTRP14 were investigated with two electron donors, dithiothreitol (DTT) and SjTGR, with SjTrx1[C34S] and SjTrx1[C37S] as controls. It could be shown that SjTrx1i could reduce insulin with both electron donors, but SjTRP14 could not (Fig 5C-5D), indicating that SjTRP14 exhibited no insulin-reducing activity, either with

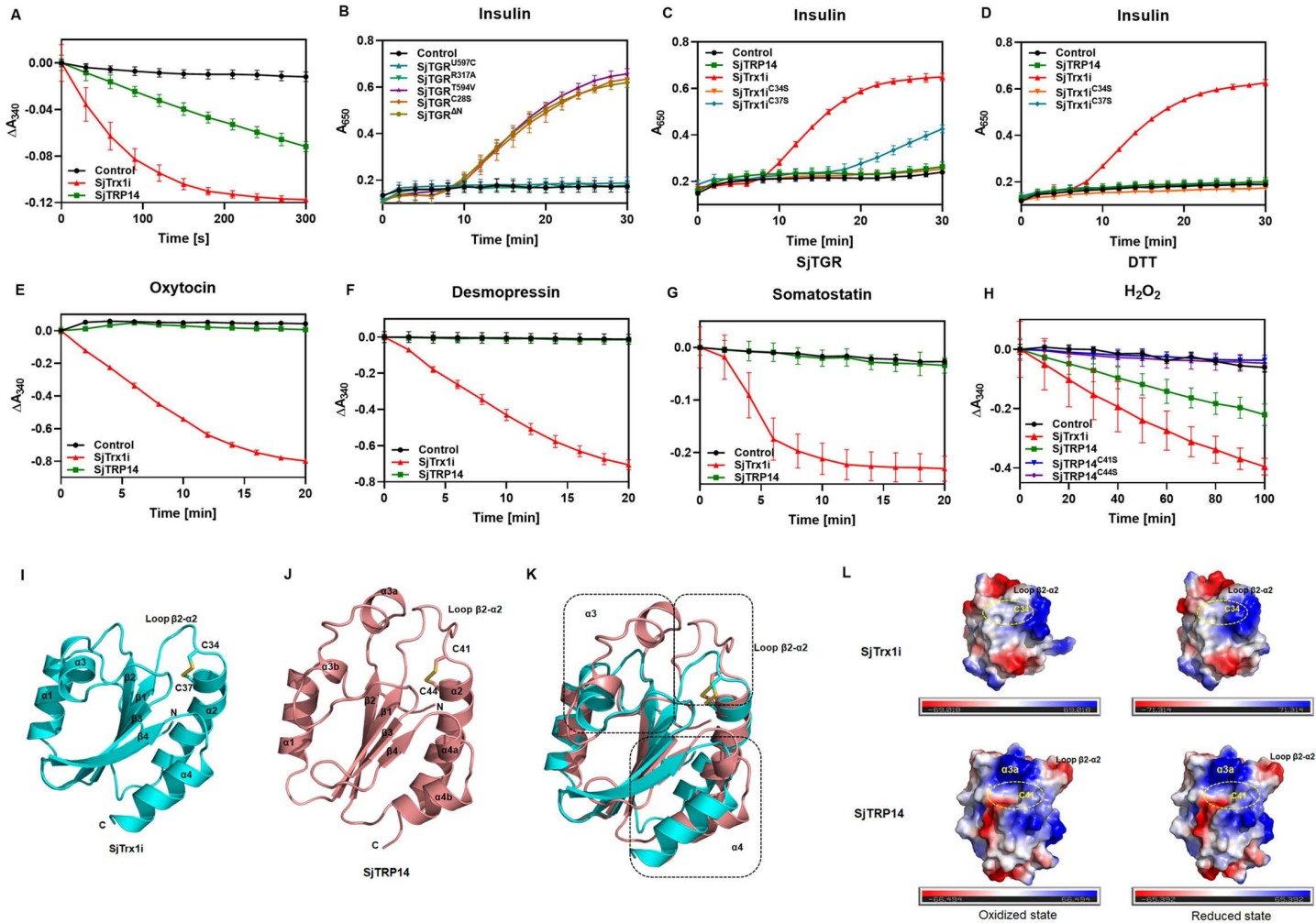

**Fig 5. Structural insights into the substrate specificity of SjTGR's downstream substrates SjTrx1i and SjTRP14.** (A) Assessment of the reduction effects of SjTGR on SjTrx1i and SjTRP14 (mean ± SD, n = 3). (B) Evaluation of the reduction capabilities of different SjTGR mutants on SjTrx1i. Insulin was used as the final substrate, and the amount of insulin precipitation was measured at 650 nm (mean ± SD, n = 3). (C) Reduction capabilities of SjTrx1i, its mutants, and SjTRP14 on insulin under DTT conditions (mean ± SD, n = 3). (D) Insulin reduction capabilities of SjTrx1i, its mutants, and SjTRP14 in the presence of SjTGR (mean ± SD, n = 3; In some cases, error bars are smaller than the symbol and thus not visible). (E-G) Assessment of the reduction capabilities of SjTrx1i and SjTRP14 on oxytocin, desmopressin, and somatostatin under SjTGR conditions (mean ± SD, n = 3; In some cases, error bars are smaller than the symbol and thus not visible). (H) Evaluation of the hydrogen peroxide ($H_2O_2$) reduction capabilities of SjTrx1i, SjTRP14, SjTRP14$^{C41S}$, and SjTRP14$^{C44S}$ under SjTGR conditions (mean ± SD, n = 3). (I) Cartoon representation of the structure of SjTrx1i (PDB: 22FJ). The disulfide bond between C34 and C37 is represented using blue sticks. (J) Cartoon depiction of the SjTRP14 (PDB: 22FG) structure, with the disulfide bond between C41 and C44 represented as salmon sticks. (K) Structural alignment of SjTrx1i (PDB: 22FJ) and SjTRP14 (PDB: 22FG). Helices α3 and α4, along with the loop β2-α2, are highlighted with a black dashed rectangle. (L) Surface charge distribution maps of SjTrx1i and SjTRP14 in their oxidized and reduced states. The enzyme active site pockets are highlighted with yellow dashed circles.

DTT or when coupled with the SjTGR enzymatic reducing system. However, SjTrx1$^{C37S}$ could still use DTT as the electron donor to maintain a partial reducing capacity. In addition, SjTrx1i could also reduce other disulfide-containing peptides including oxytocin, desmopressin, and somatostatin, but SjTRP14 did not (Fig 5E-5G). It was reported that both Trx1 and TRP14 could reduce hydrogen peroxide ($H_2O_2$) [32]. It could be demonstrated that both SjTrx1i and SjTRP14 were capable of reducing $H_2O_2$, but SjTrx1i was with much higher catalytic rate (Fig 5H). This difference may be attributed to

the slower reduction rate of SjTRP14 by SjTGR. The catalysis of SjTRP14 was also dependent on the C41-C44 motif as neither SjTRP14$^{C41S}$ nor SjTRP14$^{C44S}$ could reduce $H_2O_2$ (Fig 5H). These results suggested that SjTrx1i and SjTRP14 have different substrates specificity.

The crystal structures of SjTrx1i and SjTRP14 were determined (S6 Table). The oxidized state structure of SjTrx1 was previously reported [36]. The crystal structure of SjTrx1i was determined to a resolution of 2.66 Å in the oxidized state (PDB: 22FJ) and to a resolution of 1.90 Å in the reduced state (PDB: 22FK) (Figs 5I, S10B-S10C, S6 Table). The crystal structures of SjTRP14 were also determined for the first time to a resolution of 1.69 Å for the oxidized state (PDB: 22FG) and to a resolution of 1.77 Å (PDB: 22FH) for the reduced state (Figs 5J, S10D-S10E; S6 Table). The SjTRP14 structure adopted a typical Trx fold containing four α-helices and four β-sheets in the order of β1α1β2α2β3α3β4α4. In the oxidized state of SjTRP14, a disulfide bond between the highly conserved redox-active sites C41 and C44 was visible in the electron density map, while it was not found in its reduced state (Figs 5J, S10D-S10E). Like SjTrx1i, the highly conserved redox-active WCXXC motif of SjTRP14 was located at the N terminus helix α2 (S10A and S11 Figs). The overall structures of SjTrx1i and SjTRP14 in oxidized state were superimposed and compared for analysis of the structural differences (Fig 5K). The major differences were in helices α3 and α4, and β2-α2 loop. In contrast to that of SjTrx1i, the helices α3 and α4 of SjTRP14 appeared to be formed by two segments and could be divided into α3a-α3b and α4a-α4b. Theβ2-α2 loop was located in front of the redox-active WCPDC motif and was reported to be responsible for the substrate recognition and specificity. Theβ2-α2 loop of SjTRP14 was longer and more flexible than that of SjTrx1i, which was likely to hinder the binding of substrates, such as insulin. In addition, the extended α3a of SjTRP14 was closer to the β2-α2 loop to influence the substrate recognition. The surface characteristics and charge distribution of SjTrx1i and SjTRP14 were also analyzed and compared (Fig 5L). SjTRP14 was with deeper substrate binding pocket, suggesting that SjTRP14 might not be favorable for binding larger substrates, such as insulin. The α3a region in SjTRP14 was of positively charged close to the substrate binding pocket.

## 2.6 Knockdown of SjTGR or SjTRP14 blocked the survival of schistosomes and ameliorated egg-induced granulomatous pathology in mice

To further evaluate the impacts of the SjTGR-mediated thiol-disulfide redox signaling pathway on the survival and oviposition of schistosomes, we performed a 38-day *in vivo* RNAi on SjTGR and its downstream substrate SjTRP14 starting from the 1st day of infection, individually (Fig 6A). Compared to the control RNAi group, the expression of these two genes were significantly silenced with no sex bias (Fig 6B-6C). Meanwhile, the worm burden in the SjTGR RNAi and SjTRP14 RNAi groups were significantly reduced (Figs 6D, S12A-S12B), indicating they are essential for the schistosome's survival. Next, we also compared and analyzed the deposited eggs in mice livers and egg-induced granulomatous pathology in mice. In these two groups, the number of liver eggs were significantly decreased (Fig 6E), and the liver damage appeared weaker than the control group, as indicated by the color of the liver, the liver index, the spleen index, and the AST/ALT ratio (Figs 6F-6I, S12C-S12D). H&E staining showed that the number of egg-induced granuloma and the size of granuloma area formed by eggs in the livers of mice infected with SjTGR or SjTRP14 (RNAi) worms were significantly smaller than the control group (Fig 6J-6L). In addition, mice infected with SjTGR or SjTRP14 (RNAi) worms showed significantly decreased collagen deposition around the eggs as evaluated by Masson's trichrome staining, compared with control group (Fig 6M-6O).

These results suggested that SjTGR or its downstream substrate SjTRP14 is essential for the survival and oviposition of schistosomes, providing evidence for antiparasitic development targeting these two proteins.

## 3 Discussion

Although remarkable progresses have been made to control schistosomiasis, more effective therapeutic potions for eliminating this disease burden are still urgently needed [1,2,37]. TGR is well demonstrated to be one of the most promising drug targets and vaccine candidates against schistosome infection [16–19,38]. It is important to characterize the

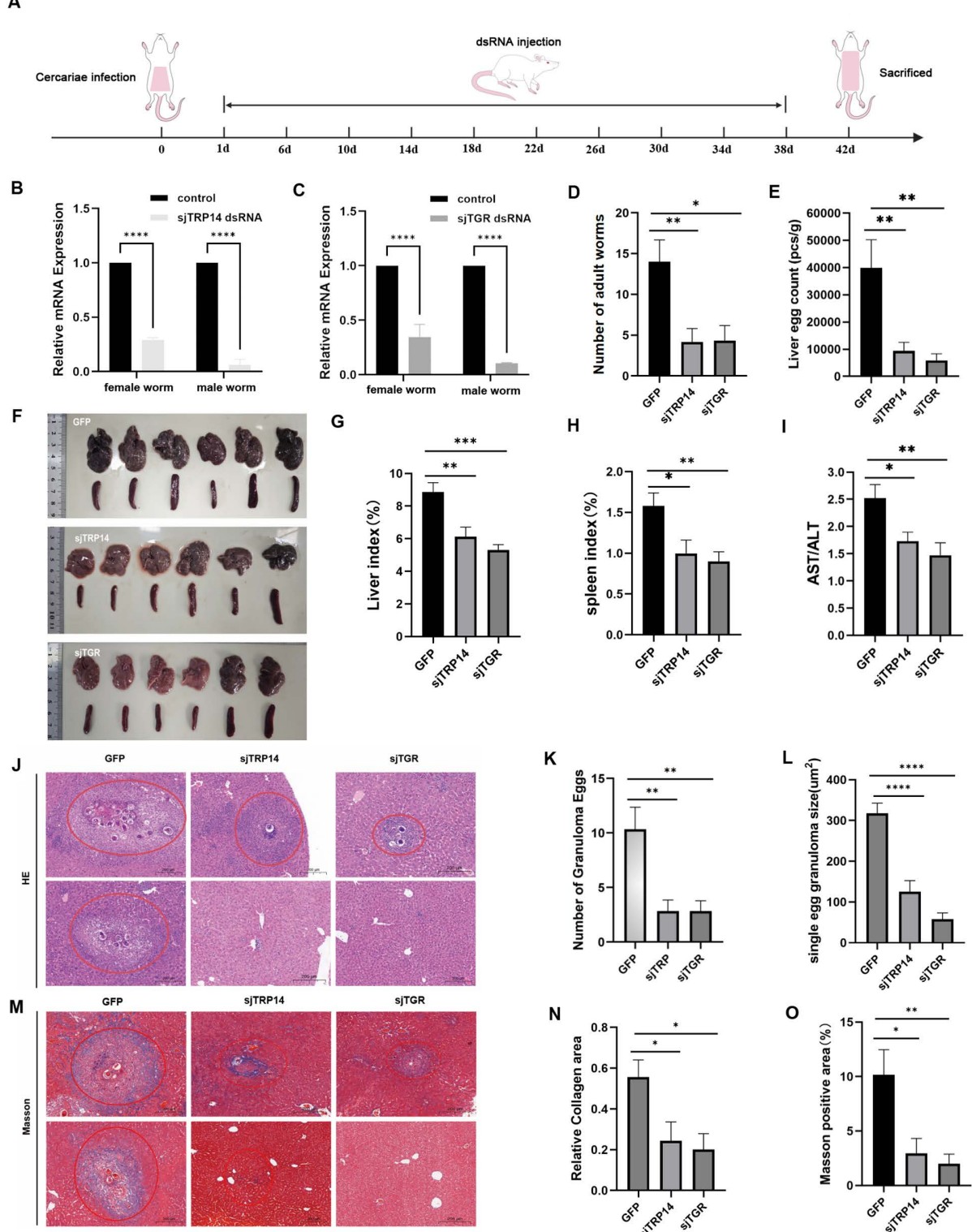

**Fig 6. Effects of *in vivo* RNAi of SjTGR and SjTRP14 on the survival of schistosome and liver granulomatous pathology.** (A) Schematic diagram depicting *in vivo* RNAi of SjTGR and SjTRP14. Infected mice were injected intravenously via the tail with 10 µg of dsRNA per injection every 4 days

starting from 1 dpi. Samples were collected at 42 dpi. The images are used under the Creative Commons CC0 1.0 Universal Public Domain Dedication. Available at: (https://w.wiki/Hubf) and (https://w.wiki/Hua6). **(B-C)** qPCR analysis of the RNAi efficacy for SjTGR and SjTRP14 in harvested adult worms (males and females separated). All experiments were performed in triplicate, and data are presented as the mean±SD. In some cases, error bars are smaller than the symbol and thus not visible. (D) Worm burden of the parasites recovered at 42 dpi in GFP (RNAi), SjTRP14 (RNAi), and SjTGR (RNAi) groups. 6 mice per group. (E) Egg count per gram of liver after RNAi. 6 livers per group. (F) Gross observations of the mouse liver after RNAi. Scale bars: 1 cm. (G-H) The liver or spleen index in mice after RNAi. (I) AST/ALT ratio in mice after RNAi. (J) Histological analysis of mouse liver by H&E (hematoxylin and eosin) staining. Scale bars: 200 µm. Images are representative of one mouse per group (n=6). (K-L) Statistical analysis of the number of egg-induced granuloma or the size of egg granuloma area in the mouse liver after RNAi. (M) Histological analysis of mouse liver by Masson's trichrome staining. Scale bars: 200 µm. Images are representative of one mouse per group (n=6). (N-O) Statistical analysis of the collagen area or the size of Masson's positive area in the mouse liver after RNAi. For all the mouse experiments, each group contained six biological replicates, and the data are presented as mean±SD. Differences are statistically significant (****$P<0.0001$, ***$P<0.001$, **$P<0.01$, *$P<0.05$).

structures of TGR and intermediates in the catalytic cycle to establish the basis for developing novel therapeutics. In this study, the entire electron transfer cycle of SjTGR was mapped by X-ray crystallography. When NADPH binds to the oxidized SjTGR, it transfers the electrons sequentially to FAD, to the nearby C154/C159 redox couple, then to C596/U597 redox active couple on the C-terminal segment of the neighboring subunit, and ultimately to either various downstream oxidized substrates such as SjTrx1 and SjTRP14 or back to the redox-active C28-C31 motif in the Grx domain of the neighboring monomer to reduce GSSG (Fig 7). These results provided valuable insights into the schistosome-host interaction driven by the redox signaling and contributed to the discovery and development of therapeutic compounds targeting SjTGR.

The structure of the active site in the C-terminus of TGR was of importance for developing therapeutics and delineating its catalysis. Yet, the electron density in this region was not well defined in previous crystal structures of TrxR1 and TGR. It was the first glimpse to the C-terminus of SjTGR, including the structure for the critical GCUG and GCCG motifs. The structures in this study unveiled the important SjTGR in the crucial $EH_4$ state for the enzymatic activity of TrxR [39,40], along with two $EH_2$ conformations and the fully oxidized conformation $E_{ox}$. The C-terminal structure adopted a cis-conformation in both the oxidized and reduced states. The hook-shaped conformation stabilized by T594-mediated hydrogen bonding was unique for SjTGR. Intriguingly, this hook-shaped conformation was exclusive to subunit B, with subunit A in disordered C-terminal structure, suggesting functional asymmetry. Furthermore, although the T594V mutation reduced the catalytic efficiency of SjTGR toward the small-molecule substrate DTNB by approximately 13% (Fig 1L-1M; Table 1), it did not impair insulin reduction in the coupled Trx1 system (Fig 5B). This divergence likely reflects distinct conformational requirements for direct substrate turnover versus protein-mediated electron transfer. The hook-shaped C-terminal conformation stabilized by T594 appears to be more critical for optimizing the geometry of direct selenothiol–substrate interactions than for maintaining productive engagement with the physiological redox partner SjTrx1i.

Mechanistically, R317 underwent redox-dependent repositioning to anchor NADPH via adenine ribose interactions, presenting a novel target for inhibitor design. The biochemical and structural data in this study demonstrated that the Grx domain, but not the N-terminal redox center, mediated the GSSG reduction, resolving conflicting models in flatworm TGR catalysis [41,42]. These findings define SjTGR's non-mammalian structural features as the prime target for species-specific antischistosomal drug development.

While prior studies identified multiple Auranofin (Au) binding sites in thioredoxin reductases (TrxRs) and TGR homologs including the N-terminal C154-C159 disulfide in SmTGR [20], C519-C573 in *Echinococcus granulosus* TGR [43], C22 in the Grx-like domain of *Brugia malayi* TrxR [44], and C189 in human TrxR1 [45], no structural evidence had linked Au binding to direct inhibition of the C-terminal redox center. This knowledge gap was bridged through the structural determination of SjTGR in complex with Au. Our findings demonstrated, for the first time, that Au simultaneously targeted three catalytic sites, which included the TrxR-domain N-terminal disulfide (C154-C159), the C-terminal selenothiol motif (C596-U597/C597), and the Grx-domain active site (C28-C31). This unique tripartite inhibition mechanism among characterized TrxR/

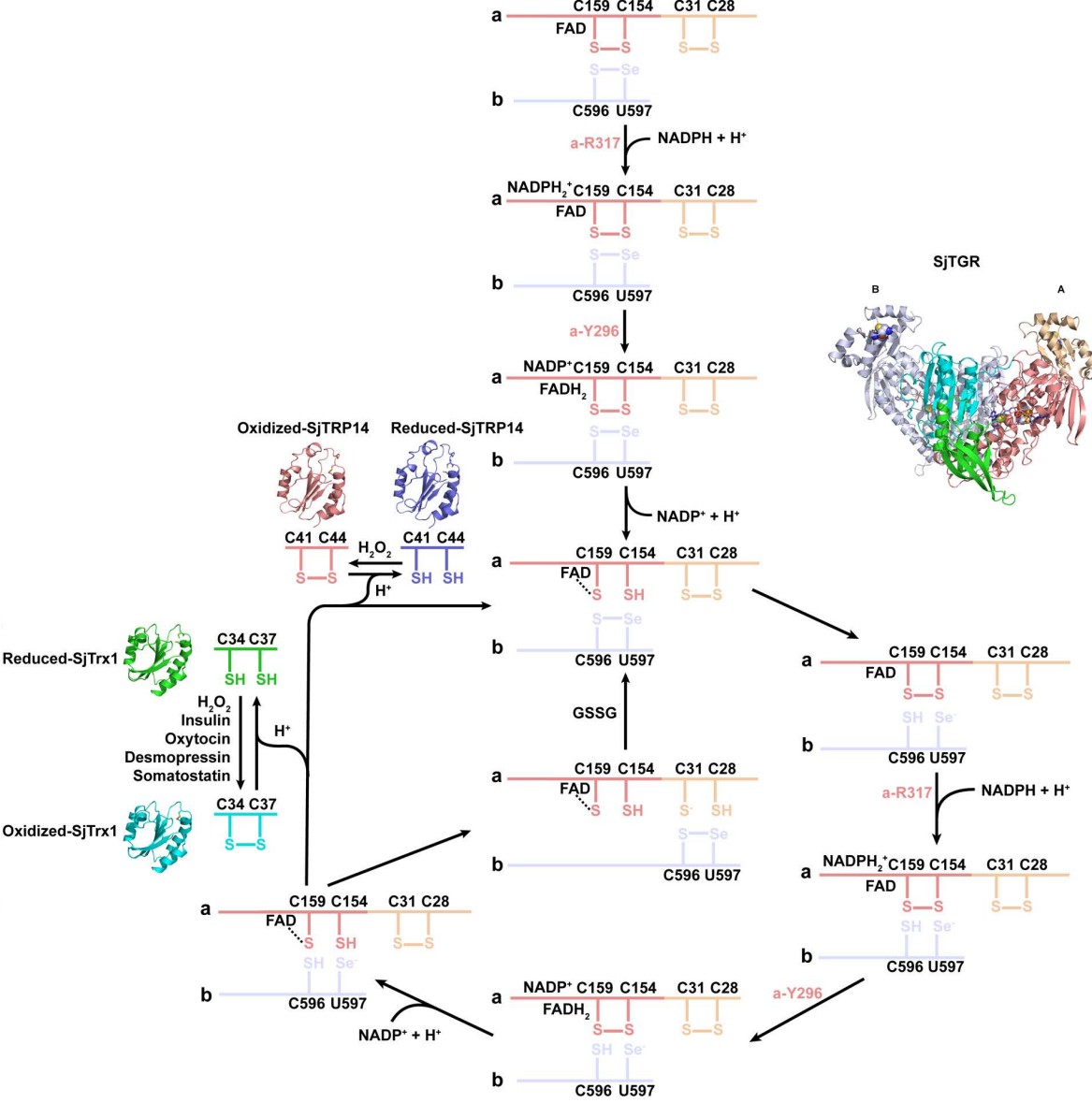

**Fig 7. Schematic diagram depicting the structural basis of electron transfer and downstream substrates enzymatic activities.** After binding to the oxidized SjTGR, NADPH transfers the electrons from its nicotinamide ring to the cofactor FAD. Then, reduced $FADH_2$ donates electrons to the nearby C154/C159 redox active couple of the enzyme. Next, electrons reach the C596/U597 redox active couple on the C-terminal segment of the neighboring subunit. Ultimately, electrons would be transferred either to various downstream oxidized substrates such as SjTrx1i and SjTRP14 or back to the redox-active C28-C31 motif in the Grx domain of the neighboring monomer to reduce GSSG.

TGR systems reveals Au's capacity to covalently block electron transfer at both interdomain (TrxR-to-Grx) and intersub-unit (C-terminal relay) levels. Crucially, it was the first structural evidence of Au binding to a C-terminal redox center in any flavoreductase, linking this interaction to the enzymatic inactivation. These findings redefined Au's inhibition paradigm in flatworm TGRs, highlighting its dual role in disrupting redox relay geometry and selenocysteine-mediated catalysis. By mapping non-mammalian Au-binding hotspots (e.g., Grx-domain C28-C31), this work established a foundation for

designing next-generation inhibitors that exploit parasite-specific structural vulnerabilities, advancing efforts to combat auranofin resistance and improve schistosomiasis therapeutics.

Additionally, both the irreversible covalent inhibitor TRi-1 and the reversible inhibitor Sj001 also demonstrated significant inhibitory activity against SjTGR in enzymatic assays. However, despite repeated attempts using crystal soaking and co-crystallization, we were unable to resolve high-resolution structures of SjTGR in complex with either TRi-1 or Sj001. This difficulty may arise from crystallographic technical challenges, such as ligand-induced crystal disorder, conformational heterogeneity, or crystallization condition incompatibility, and does not necessarily imply weak binding affinity. Furthermore, as standard binding affinity assays rely on reversible association-dissociation equilibria [10], reliable quantification of Auranofin and TRi-1 binding affinities for SjTGR was technically unfeasible. In contrast, Sj001 acts as a reversible inhibitor of SjTGR, enabling valid measurement of its equilibrium binding affinity, this is why only Sj001 binding data are presented in the Fig 4H.

As key members of the thioredoxin family, Trx1 and TRP14 are regulated by TrxR1 and fulfill distinct biological roles within cellular redox systems [12,44–46]. In our *in vitro* assays, we demonstrated that SjTGR effectively reduced both SjTrx1i and SjTRP14, while exhibiting substrate-specific selectivity. Both proteins mediate the reduction of hydrogen peroxide while SjTRP14 lacked the catalytic efficiency exhibited by SjTrx1i towards disulfide-containing peptides, including insulin, oxytocin, desmopressin, and somatostatin. Concerning the inclusion of SjTRP14$^{C41S}$ and SjTRP14$^{C44S}$ exclusively in Fig 5H, wild-type SjTRP14 exhibited no detectable catalytic activity in the assays shown in Fig 5E-5G. We therefore omitted these two cysteine mutants from these panels, as no functional readout was available to assess mutational effects. In contrast, wild-type SjTRP14 displayed clear peroxidase activity toward hydrogen peroxide in Fig 5H, prompting us to include these mutants to determine whether the Cys41 and Cys44 residues are essential for this catalytic function. Despite their structural similarities, key differences existed between the two. Notably, SjTRP14 featured an elongated β2-α2 loop compared to SjTrx1i, and while SjTrx1i possessed a continuous helix α3, SjTRP14's helix α3 was divided into two distinct segments, α3a and α3b. The extended β2-α2 loop in SjTRP14 likely induced steric hindrance, while the positively charged region near its catalytic pocket formed by α3a could influence substrate recognition and catalytic activity. These structural distinctions may account for the observed differences in substrate specificity between SjTrx1i and SjTRP14. Further investigation is required to fully elucidate the underlying mechanisms driving these functional divergences.

Although several studies have demonstrated the critical function of SjTGR in schistosome's growth and development [28–30], *in vivo* RNAi was firstly performed in our study to indicate that SjTGR and its downstream substrate SjTRP14 plays an essential role in the survival and oviposition of schistosomes, thus ameliorating egg-induced granulomatous pathology in mice. The severe phenotypes resulting from SjTGR knockdown including reduced worm burden and suppressed fecundity, closely mirror the established anti-schistosomal effects of pharmacological TGR inhibitors, such as auranofin [17,18,20] and other inhibitors [10,21–23,31]. This functional congruence strongly reinforces TGR as a critical and druggable hub within the parasite's redox defense system. In contrast, the similarly profound impairment caused by SjTRP14 silencing has not, to our knowledge, been previously recapitulated by chemical inhibition, as specific TRP14 inhibitors remain undeveloped. Our study therefore provides the first direct genetic evidence that TRP14 is essential for *S. japonicum* survival and reproduction *in vivo*, nominating it as a compelling and novel target for future therapeutic discovery aimed at disrupting the parasite's thioredoxin pathway. However, the mechanisms involved in this process still require further studies. These results showed that interference of the SjTGR-mediated thiol-disulfide redox signaling pathway will contribute the development of antischistosomal drugs.

Collectively, the entire catalytic cycle of SjTGR was mapped by X-ray crystallography and the structural basis of its catalytic activity was determined. Moreover, this study provided structural insights into the substrate specificity between SjTrx1i and SjTRP14, which could be regarded as the physiological substrates of SjTGR. This study enriched the current understanding of the thioredoxin family and their differential roles in redox regulation. More importantly, these results also provided several novel regulatory mechanisms involved in redox homeostasis maintained by SjTGR, thereby contributing

PLOS Pathogens

to the discovery and development of more effective antischistosomal agents targeting SjTGR for treating schistosomiasis and the global elimination of this disease burden.

## 4 Materials and methods

### 4.1 Ethics statement

All animal experiments were approved by the Ethics Committee of School of Basic Medical Science, Central South University (approval ID: 2021-KT25).

### 4.2 Materials

The plasmid pABC2a was generously provided by Prof. Elias Arnér and Dr. Qing Cheng (Karolinska Institutet, Sweden). The *E. coli* strain C321.ΔA was obtained from Addgene (Watertown, MA, USA), while DH5α and BL21 (DE3) competent cells were purchased from AngYuBio Co., Ltd. (Shanghai, China). Sodium selenite, ampicillin sodium salt, kanamycin sulfate, and IPTG were sourced from Adamas-beta (Shanghai, China) and INALCO Pharmaceuticals (Carlsbad, CA, USA), respectively. NADPH (β-Nicotinamide adenine dinucleotide phosphate tetrasodium salt), carbenicillin disodium salt, glutathione (reduced and oxidized forms), oxytocin, and auranofin were acquired from Biosharp (Anhui, China), Sangon Biotech (Shanghai, China), and MedChemExpress (Monmouth Junction, NJ, USA). Streptomycin sulfate, recombinant human insulin, desmopressin acetate, and somatostatin acetate were procured from Beyotime Biotechnology (Shanghai, China).

### 4.3 Parasites and animals

Cercariae of *S. japonicum* required for animal infection in this study were released from infected *Oncomelania hupensis* snails, provided by the Jiangsu Institute of Parasitic Diseases (JIPD). Female SPF-class KunMing mice, aged 6 weeks, were purchased from Department of Laboratory Animals, Central South University. For the *in vivo* experiments, each mouse was infected with $30 \pm 2$ cercariae. Worms were collected from infected mice at 42 dpi by perfusion using a saline solution containing sodium heparin. All animal experiments were approved by the Ethics Committee of School of Basic Medical Science, Central South University (approval ID: 2021-KT25).

### 4.4 Cloning, expression, and purification of SjTrx1i, SjTRP14, SjTGR and its mutants

We constructed the plasmids capable of expressing selenoproteins according to the method described by Qing Cheng et al., which uses a predefined UAG codon to encode selenocysteine (Sec), the pABC2a vector (encoding mutated SelC that can complement UAG, as well as SelA and SelB), and the RF1-depleted *E. coli* strain C321.ΔA [47,48]. Notably, this expression system can achieve close to 100% Sec incorporation if supplemented with the SECIS element (Selenocysteine Insertion Sequence) [47,48].

The cDNAs encoding SjTGR[WT] and SjTGR[ΔN] were cloned into the N-terminal His-tagged pABC2a vector, with the *E. coli*-derived SECIS element (GGTTGCAGGTCTGCACC) added after the GCUG sequence [47,48]. These recombinant plasmids were subsequently transformed into *E. coli* strain C321.ΔA competent cells for the expression of selenocysteine-containing proteins. The SjTGR[U597C] and SjTrx1i genes were cloned into the pET-28a vector with an N-terminal His-tag, whereas the SjTRP14 gene was inserted into the pET-22b vector with a C-terminal His-tag, and all recombinant proteins were expressed in *E. coli* BL21 (DE3) cells. All mutants, including SjTGR[T594V], SjTGR[C28S], SjTGR[R317A], SjTrx1i[C34S], SjTrx-1i[C37S], SjTRP14[C41S], and SjTRP14[C44S], were generated through site-directed mutagenesis employing the overlap extension PCR technique. The primers used in this study were listed in S2 Table.

Selenoprotein expression was carried out as previously described with LB medium adaptation for SjTGR[WT] and SjTGR[ΔN] [47,48]. The *E. coli* C321.ΔA cultures were grown at 30°C until reaching an $OD_{600}$ of 0.8-1.0, followed by the addition of 5 μM sodium selenite and 0.2 mM IPTG at 16°C for 16 hours for protein expression and selenocysteine incorporation.

Expression plasmids encoding non-selenocysteine-containing proteins including SjTGR$^{U597C}$, SjTrx1i and SjTRP14 were transformed into *E. coli* BL21 (DE3) cells. These cultures were incubated at 37°C with agitation until reaching an OD$_{600}$ of 0.8-1.0 for the induction of protein expression with 0.2 mM IPTG at 16°C for 16 hours.

The bacterial pellets were resuspended in a buffer containing 20 mM Tris, 300 mM NaCl, pH 8.0 and the resuspended cells were subjected to ultrasonication, followed by centrifugation. The supernatant was then loaded onto a Ni-affinity chromatography column. The column was washed with a buffer containing 25 mM imidazole, and the target protein was eluted using a buffer containing 250 mM imidazole. Following affinity chromatography, further purification was carried out with a Superdex 200 Increase 10/300 GL size-exclusion chromatography column. The molecular weight and purity of the proteins were subsequently evaluated by SDS-PAGE analysis. The mutants of SjTGR, SjTrx1i, and SjTRP14 were expressed and purified similarly.

### 4.5 Determination of selenium incorporation efficiency

The Selenium (Se) incorporation efficiency of the SjTGR$^{WT}$ protein sample (1 µM) was determined by Inductively Coupled Plasma Mass Spectrometry (ICP-MS). Briefly, a 5 mL aliquot of the sample was digested with concentrated HNO$_3$ using a programmed microwave digestion system and subsequently diluted to a final volume of 25 mL with deionized water. Total Se content was quantified using an Agilent 7800 ICP-MS system equipped with a concentric nebulizer and a spray chamber maintained at 2°C. The instrument was operated at an RF power of 1550 W with a plasma gas flow of 15 L/min and an auxiliary gas flow of 0.9 L/min. To minimize polyatomic interferences, the collision cell was utilized in Helium mode with a flow rate of 5 mL/min. External calibration was performed using a series of Se standards ranging from 0 to 1000 µg/L. Each measurement was conducted in triplicate with an integration time of 0.3 s per mass to ensure statistical reliability, and the analysis confirmed a selenocysteine incorporation efficiency of 92.3% (the mean of triplicate measurements) (S7 Table).

### 4.6 Crystallization of SjTGR, SjTrx1i and SjTRP14

The purified proteins were concentrated to a final concentration of 10 mg/ml in a buffer consisting of 20 mM Tris, 200 mM NaCl at pH 8.0, for crystallization. The crystals were grown using the hanging-drop vapor diffusion technique.

Both SjTGR$^{WT}$ and SjTGR$^{U597C}$ crystals were grown at 22°C in a reservoir solution of 4% Tacsimate (pH 6.5-8.0) and 15–20% PEG3350, and the crystals appeared in 2–7 days. To form the complex, SjTGR$^{U597C}$ protein was incubated with NADPH, GSH, and Auranofin separately at a molar ratio of 1:3 for 6 hours. Crystallization trials were carried out under the same reservoir condition for the apo proteins. The crystals were cryoprotected by transient immersion in the reservoir solution supplemented with 25% (v/v) glycerol and flash-cooled in liquid nitrogen at 72 h post-crystallization for the X-ray diffraction analysis at Shanghai Synchrotron Radiation Facility (SSRF).

SjTrx1i crystals were obtained by mixing 1 µL of SjTrx1i protein with 1 µL of reservoir solution containing 100 mM SPG buffer (pH 4.5) and 25% PEG1500, followed by incubation at 22°C for approximately 7 days. In contrast, SjTRP14 crystals were grown at 16°C in a reservoir solution containing 0.2 M lithium sulfate monohydrate, 0.1 M HEPES (pH 7.5), and 25% PEG3350, with a similar growth period of about 7 days. To determine the reduced structures of SjTrx1i and SjTRP14, the crystals were soaked in 5 mM DTT for 10 minutes and then immediately flash-frozen in liquid nitrogen.

### 4.7 Data collection, structure determination, and refinement

X-ray diffraction data were collected at 100 K using beamlines BL-02U1, BL-10U2, and BL-19U1 at the SSRF in China, and processed with XDS [49]. Structural models of SjTGR, SjTrx1i, and SjTRP14 were generated using AlphaFold2 for the structure determined by molecular replacement with PHASER [50]. The model building was conducted using Coot [51], followed by iterative cycles of crystallographic refinement using REFMAC5 [52]. Water molecules were added both automatically and manually with Coot. The statistics for data processing and structure refinement are summarized in the Supplementary Tables 2, 3 and 5. Structural figures were generated using PyMOL (http://www.pymol.org/).

## 4.8 Inhibitors synthesis

### 4.8.1 Synthesis of ethyl (E)-3-(6-(trifluoromethyl)pyridin-3-yl)acrylate.

To a stirred solution of triethyl phosphonoacetate (3.1 mmol) in THF (10 mL) was added NaH (3.3 mmol) at 0°C. The mixture was stirred at the same temperature for 1.5 h. and then it was stirred at room temperature for 1 h. Nicotinaldehyde (2.9 mmol) was added slowly to the reaction mixture and was stirred at room temperature overnight. The reaction was quenched with 30 mL brine and the THF was removed under vacuum. The water phase was extracted with ethyl acetate and dried with anhydrous $Na_2SO_4$. The mixture was filtered and ethyl acetate was removed under vacuum. The concentrate was purified by chromatography on a silica gel column to give 2, colorless oil, 0.47 g, yield 67%. 1H NMR (600 MHz, CHLOROFORM-d) ppm 1.32 - 1.41 (m, 3 H) 4.31 (q, J = 7.15 Hz, 2 H) 6.60 (d, J = 16.14 Hz, 1 H) 7.68 - 7.77 (m, 2 H) 8.01 (d, J = 8.07 Hz, 1 H) 8.86 (s, 1 H). LCMS m/z: 246.2 [M + H]+.

### 4.8.2 Synthesis of (E)-3-(6-(trifluoromethyl)pyridin-3-yl)prop-2-en-1-ol.

To a solution of 2 (1.9 mmol) in $CH_2Cl_2$ (5 mL), DIBAL-H was added dropwise (1 M in toluene, 5.8 mmol) at -78°C over 10 min. The reaction mixture was stirred at the same temperature for 4 h. The reaction was quenched with MeOH and the mixture was stirred for 0.5 additional hour. The flask was moved to the ambient atmosphere and the aqueous solution was added with excess sodium potassium tartrate. The mixture was stirred at room temperature for 4 h. The organic layer was separated and the aqueous layer was extracted with ethyl acetate. The combined organic layer was washed with brine and water, dried with anhydrous $Na_2SO_4$, filtered with celite and concentrated under vacuum. The crude product was purified by chromatography on a silica gel column to give 3, yellow oil, 0.34 g, yield 87%. 1H NMR (600 MHz, CHLOROFORM-d) ppm 4.42 (d, J = 4.77 Hz, 2 H) 6.47 - 6.61 (m, 1 H) 6.71 (d, J = 16.14 Hz, 1 H) 7.65 (d, J = 8.07 Hz, 1 H) 7.86 (d, J = 8.07 Hz, 1 H) 8.72 (s, 1 H). LCMS m/z: 204.1 [M + H]+.

### 4.8.3 Synthesis of 3-(hydroxymethyl)-4-(6-(trifluoromethyl)pyridine-3-yl)-1,2,5-oxadiazole 2-oxide Sj001.

To a solution of 3 (0.5 mmol) in AcOH (1.5 mL), sodium nitrite (0.4 mmol x 6) was added portion-wise over 2 h. The reaction was stirred at 60°C overnight. The solvent was removed under vacuum and the residue was dissolved in ethyl acetate. The organic layer was washed with brine and water, dried with anhydrous $Na_2SO_4$, filtered and concentrated under vacuum to afford crude product which was purified by chromatography on a silica gel column to give Sj001, yellow oil, 30 mg, yield 23%. 1H NMR (600 MHz, CHLOROFORM-d) ppm 4.42 2.67 (br. s., 1 H) 4.80 (br. s., 2 H) 7.91 (d, J = 8.25 Hz, 1 H) 8.46 (br. s., 1 H) 9.27 (s, 6 H). LCMS m/z: 262.2 [M + H]+.

### 4.8.4 Synthesis of 2-((4-chlorophenyl)sulfonyl)-6-methoxy-3-nitropyridine TRi-1.

To a solution of 2-chloro-6-methoxy-3-nitropyridine (1.1 mmol) in DMF (3 mL), sodium 4-chlorobenzenesulfinate (1.6 mmol), tetrabutyl ammonium chloride (0.3 mmol), and concentrated HCl (0.03 mL) was added sequentially. The reaction mixture was stirred at 80°C for 1.5 h and quenched with ice water and filtered. The filtrate was washed with water and dried to give TRi-1, gray solid, 0.33 g, yield 95%. 1H NMR (600 MHz, DMSO-d6) ppm 3.63 (s, 3 H) 7.30 (d, J = 8.80 Hz, 1 H) 7.80 (d, J = 8.62 Hz, 2 H) 8.02 (d, J = 8.62 Hz, 2 H) 8.50 (d, J = 8.80 Hz, 1 H). LCMS m/z: 329.7 [M + H]+.

## 4.9 Enzyme activity assays

### 4.9.1 DTNB assay.

The DTNB (5,5'-dithiobis(2-nitrobenzoic acid)) reductase activity assay was performed in 100 mM HEPES (pH 7.0) containing 1 mM EDTA, 600 µM NADPH, and 50 nM recombinant SjTGR (wild-type or mutants) using a Tecan Spark microplate reader. Reactions were initiated by rapid injection of 1 mM DTNB, with absorbance at 412 nm ($\varepsilon$ = 13.6 mM$^{-1}$·cm$^{-1}$) recorded every 30 s for 10 min under isothermal conditions (25.0 ± 0.2°C). For reversible inhibition assessment, SjTGR was pre-incubated with 100 µM inhibitor Sj001 (2 h, 4°C) prior to assay; subsequent dialysis against 20 mM Tris-HCl (pH 8.0) supplemented with 200 mM NaCl was conducted twice (24 h each, 4°C) utilizing 10 kDa MWCO membranes to remove ligands, followed by determining the enzymatic activity with SjTGR + 1% (v/v) DMSO as the control.

The catalytic activity toward DTNB was assayed in reaction systems containing 50 nM SjTGR^WT or SjTGR^V594T, 400 µM NADPH, and DTNB gradients (20–500 µM) in 100 mM HEPES buffer (pH 7.0) supplemented with 1 mM EDTA. Initial velocities derived from $\Delta A_{412}$ linear-phase slopes (0–60 s) were plotted against the substrate concentrations. Kinetic

parameters were determined by fitting the data to the Michaelis-Menten nonlinear equation using nonlinear regression analysis (GraphPad Prism 8.0).

**4.9.2 Assaying GSSG and HEDS.** For the GSSG and HEDS assays, the decrease in NADPH concentration was measured. Specifically, the GSSG assay utilized a reaction mixture comprising 100 mM HEPES buffer (maintained at pH 7.0), 1 mM EDTA, 600 µM NADPH, 50 nM SjTGR or its mutants, and 1 mM GSSG. The HEDS assay reaction mixture consisted of 100 mM HEPES buffer (pH 7.5), 1 mM EDTA, 600 µM NADPH, 50 nM SjTGR or its mutants, 6 µg/mL glutathione reductase, 1 mM GSH, and 1 mM HEDS, as previously reported [53]. Both assays were conducted at a temperature of 25°C, and the absorbance at 340 nm ($A_{340}$) was measured at 2 minute intervals over a 20-minute time period after the addition of the substrate.

**4.9.3 Reduction *of* SjTrx1i and SjTRP14 *by* SjTGR.** The reduction of SjTrx1i and SjTRP14 mediated by SjTGR was evaluated using a Tecan Infinite E Plex microplate reader. The reaction mixture included 100 mM HEPES (pH 7.0), 1 mM EDTA, 600 µM NADPH, 50 nM SjTGR, and 100 µM SjTrx1i or SjTRP14. $A_{340}$ was taken at 30 second intervals for a total of 300 seconds at 25°C.

**4.9.4 Insulin assay.** Insulin detection was carried out by monitoring $A_{650}$. The reaction mixture contained 100 mM HEPES (pH 7.0), 1 mM EDTA, and either 1 mM DTT or 50 nM SjTGR (or its mutants) supplemented with 600 µM NADPH as the electron donor for reducing 5 µM SjTrx1i and SjTRP14. The insulin concentration was 155 µM. The absorbance was measured at 2-minute intervals over 30 minutes.

**4.9.5 Oxytocin, desmopressin, and somatostatin reduction assay.** The reaction system for oxytocin, desmopressin, and somatostatin was 100 mM HEPES (pH 7.0), 1 mM EDTA, 600 µM NADPH, 50 nM SjTGR, and 5 µM SjTrx1i or SjTRP14. Oxytocin and desmopressin were added at a concentration of 1 mM, whereas somatostatin was used at 200 µM. $A_{340}$ was monitored every 2 minutes for 20 minutes.

**4.8.6 Hydrogen peroxide consumption assay.** For the hydrogen peroxide consumption assay, the reaction system was identical to that for oxytocin, except the concentration for hydrogen peroxide was 1 mM. Given the slow reaction rate, $A_{340}$ was taken every at 10 minute intervals at 25°C for 100 minutes.

## 4.10 Inhibition of SjTGR

Inhibition of SjTGR was assayed with Auranofin, TRi-1, and Sj001 at concentrations ranging from 1 nM to 1 µM. The reaction mixture consisted of 100 mM HEPES (pH 7.0), 1 mM EDTA, 600 µM NADPH, 50 nM SjTGR, and 1 mM DTNB. The assay was carried out at 25°C with $A_{412}$ monitored every 30 seconds over 5 minutes. A control without the inhibitors was also prepared for comparison. The rate of TNB formation was calculated by fitting linear equations to the data using GraphPad software, and the obtained results were compared with those from the control.

## 4.11 ITC assays

The isothermal titration calorimetry (ITC) experiments were carried out using a MicroCal PEAQ-ITC instrument at 25°C. The titration protocol involved one initial injection of 0.4 µL, followed by 19 injections of 2 µL, with a stirring rate of 750 rpm and a high feedback model. The buffer for the proteins were 20 mM Tris, pH 8.0, and 200 mM NaCl. For the NADPH titration of SjTGR$^{WT}$ or SjTGR$^{R317A}$, the NADPH concentration was 400 µM, and protein concentration was 20 µM. Sj001 at 1 mM and 20 µM SjTGR were used in the titration experiments. The titration data were analyzed using MicroCal PEAQ-ITC Analysis Software with a one-site binding model.

## 4.12 dsRNA synthesis

dsRNA was synthesized based on an established method previously described [54,55]. The target sequence primer was designed at the 5'end of the gene, with a length of 255 bp and 437 bp for *sjtrp14* and *sjtgr*, respectively. The primers used to synthesize dsRNA included a T7 sequence (TAATACGACTCACTATAGGGAGA) at the 5'end of each oligo, and the

dsRNA primer sequences for SjTGR and SjTRP14 used in this study are shown in S2 Table. DNA templates were amplified from cDNA of adult worms and confirmed by Sanger Sequencing. Following the manufacturer's instructions, dsRNA was synthesized using the dsRNA synthesis kit (Vazyme Biotech Co., Ltd). Finally, the dsRNA was aliquot and store at -20°C for later use.

### 4.13 *In vivo* RNAi

Following infection with 30±2 cercariae, 18 infected female Kunming mice were randomly divided into three groups (n=6 per group): the normal control group (GFP dsRNA), sjTRP14 group (sjTRP14 dsRNA), and sjTGR group (sjTGR dsRNA). Subsequently, each mouse was administered 10 µg of dsRNA (dissolved in 200 µL physiological saline) via intravenous tail vein injection on the following days post-infection (dpi): 1, 6, 10, 14, 18, 22, 26, 30, 34, and 38, respectively. This dose and schedule were selected based on preliminary studies [54–55] to effectively inhibit target gene expression in *S. japonicum* for the duration of the treatment period. Samples were collected from all the experimental mice at 42 dpi. Control mice were injected with dsRNA of *gfp*, a green fluorescent protein from *Aequorea victoria* that is not present in *S. japonicum* [55].

### 4.14 Sample collection

The body weight of all mice was measured once every 5 days. For each group, six mice at 42 dpi were sacrificed and blood samples were collected to the serum for aspartate aminotransferase (AST) and alanine aminotransferase (ALT) analysis. Whole liver and spleen samples were collected to measure weight for liver and spleen index calculation: liver-spleen index=liver or spleen weight (g)/ mouse body weight (g). The liver of each mouse was fixed with 4% paraformaldehyde for histopathological analysis.

### 4.15 Histopathological analysis

The fixed liver was stained with hematoxylin and eosin (H&E) to assess granulomas formation. And the degree of hepatic fibrosis was assessed by Masson's trichrome staining. Images of the stained sections were collected under an optical microscope and analyzed. Data analysis was performed using Image-Pro Plus software to quantitatively analyze the positive areas, further evaluating the effects of gene knockdown on hepatic lesions and fibrosis.

### 4.16 Liver egg counting

The mouse liver weighed 0.2 g was collected from the same region of all the mice and 10 mL of 5% potassium hydroxide was added. The mixture was digested on a shaker at 37°C and 220 rpm for approximately 1 hour to obtain a homogeneous solution. Next, 100 µL of the above solution was observed and counted under an ordinary upright microscope. Each sample was counted 5 times, and the average (AE) was calculated. The number of eggs per gram (EPG) of liver was calculated using the formula: EPG= (AE × 100)/ 0.2.

### 4.17 Quantitative real-time PCR analysis

Total RNA was extracted from two pairs of harvested worms (males and females separated) using a Trizol reagent according to the manufacturer's guidelines, and then analyzed by qPCR to determine the level of target gene knockdown. Subsequently, all the collected RNA were reverse transcribed to cDNAs using Revert Aid Reverse Transcriptase. qPCR analysis was performed in three technical replicates using specifically designed primers. The qPCR primer sequences for all the genes used in this study are designed and shown in S2 Table. SYBR Green using Master Mix on a CFX96 Touch System (BioRad) was used for the qPCR, and the expression level of the target gene was analyzed using the $2^{-\Delta\Delta Ct}$ method. α-Tubulin was used as the internal reference gene to calculate the relative expression level of the target gene.

## 4.18 Statistical analysis

Statistical analysis was performed with GraphPad Prism 10.4.1 for Windows (GraphPad Prism Software, San Diego, CA, USA). Experimental data were presented as mean±standard deviation (SD) for at least three independent experiments, and between-group comparisons were performed using t-tests, with $P<0.05$ indicating significant differences. All enzymatic and substrate assays were performed in triplicate. For all the mouse experiments, each group contained six biological replicates.

## Supporting information

**S1 Fig. Identification, expression and crystallization of the protein SjTGR.** (A) Sequence alignment of thioredoxin reductases from *Homo sapiens* (hTrxR1, NCBI accession: NP_877393.1), *Rattus norvegicus* (RnTrxR1, NCBI accession: O89049.5), *Schistosoma mansoni* (SmTGR, NCBI accession: XP_018649018.1), and *Schistosoma japonicum* (SjTGR, NCBI accession: ACH86016.1). (B) 15% SDS-PAGE analysis of SjTGR^U597C purification by the Ni-NTA affinity chromatography (M: marker, P: precipitate, S: supernatant, FT: flow-through, W1–4: wash fractions 1–4, E: eluate). (C) Elution profile of SjTGR^U597C from Superdex 200 Increase 10/300 GL size-exclusion chromatography. (D) Gel filtration chromatography reveals the purified SjTGR^U597C protein, which was initially obtained through Ni beads affinity chromatography and subsequently purified further using Superdex 200 size-exclusion filtration. (E) Crystals of the SjTGR^U597C protein. (F) 15% SDS-PAGE analysis of SjTGR^WT purification by the Ni-NTA affinity chromatography (M: marker, P: precipitate, S: supernatant, FT: flow-through, W1–4: wash fractions 1–4, E: eluate). (G) Elution profile of SjTGR^WT from Superdex 200 Increase 10/300 GL size-exclusion chromatography. (H) Gel filtration chromatography reveals the purified SjTGR^WT protein, which was initially obtained through Ni beads affinity chromatography and subsequently purified further using Superdex 200 size-exclusion filtration. (E) Crystals of the SjTGR^WT protein.
(TIF)

**S2 Fig. Structural superposition and comparative analysis of SjTGR and SmTGR. (A)** The FAD *Fo-Fc* electron density map for subunit A and subunit B of SjTGR (PDB: 9LWM), contoured at 3σ and displayed in green meshes. **(B)** The anomalous scattering electron density of selenium (Se), contoured at 5σ in violet meshes (PDB: 9LWM and 22EY). **(C)** Structural superposition of SmTGR (blue) (PDB: 7B02) and SjTGR (PDB: 9LWM) TrxR domains was performed, with both TrxR domains rendered transparent to accentuate divergent orientations of their Grx domains. **(D)** Electron transfer path between the Grx domain redox center (C28-C31) and the adjacent subunit's C-terminal selenothiol motif (C596-U597) in SmTGR (PDB: 7B02) and SjTGR (PDB: 9LWM). Distance measurements are shown as black dashed lines.
(TIF)

**S3 Fig. Electron transfer in SjTGR. (A-D)** The *Fo-Fc* electron density maps for the three redox centers of SjTGR-WT (GCUG) and SjTGR-U597C (GCCG) with different redox states are shown, contoured at 3σ in green meshes. The corresponding PDB entries are 9LWZ (A), 22FC (B), 22EY (C), and 9LWM (D). **(E-F)** The conformations of the last five residues (green sticks) of RnTrxR1 in the oxidized state (E) (PDB: 3EAO) and the reduced state (F) (PDB: 3EAN). **(G)** The conformations of the residues (green sticks) at the C-terminal of SmTGR (PDB: 7B02) in the reduced state. **(H-I)** The 2*Fo-Fc* electron density maps (1σ, blue meshes) and the *Fo-Fc* electron density maps (3σ, green meshes) of SjTGR subunit B C-terminal in the oxidized state (PDB: 22FC) and the reduced state (PDB: 22EY). **(J)** The 2*Fo-Fc* electron density map (1σ, blue meshes) and the *Fo-Fc* electron density map (3σ, green meshes) of the C-terminal of SjTGR (PDB: 22FC) subunit A.
(TIF)

**S4 Fig. The *Fo-Fc* electron density maps for NADPH (violet sticks).** Y296 and R317 (green sticks) within SjTGR are shown, contoured at 3σ in green meshes. In panel **(A)**, the NADPH-bound conformation of SjTGR (PDB: 22FD) is illustrated, whereas panel **(B)** displays the unbound state (PDB: 22EY).
(TIF)

**S5 Fig. ITC measurements of NADPH binding to SjTGR and SjTGR<sup>R317A</sup>.** **(A)** The two additional technical replicates for ITC analysis of NADPH binding to wild-type SjTGR. **(B)** The two additional technical replicates for ITC analysis of NADPH binding to SjTGR$^{R317A}$.
(TIF)

**S6 Fig. The conformations of the C-terminal of SjTGR under the influence of Au. (A-B)** The 2*Fo-Fc* electron density map (1σ, blue meshes) and the *Fo-Fc* electron density map (3σ, green meshes) of the C-terminal of SjTGR (PDB: 22FD) subunit A. **(C-D)** The 2*Fo-Fc* electron density map (1σ, blue meshes) and the *Fo-Fc* electron density map (3σ, green meshes) of the C-terminal of SjTGR (PDB: 22FD) subunit B.
(TIF)

**S7 Fig. ¹H NMR Spectrum of compounds and the inhibition analysis of SjTGR by Sj001. (A)** ¹H NMR Spectrum of compound Sj001. **(B)** ¹H NMR Spectrum of compound TRi-1. **(C)** Detection diagram of the reversible inhibition of SjTGR by Sj001 (mean±SD, n=3; In some cases, error bars are smaller than the symbol and thus not visible).
(TIF)

**S8 Fig. ITC analysis of Sj001 binding to SjTGR at 25°C.** The two additional technical replicates for ITC analysis of Sj001 binding to wild-type SjTGR.
(TIF)

**S9 Fig. Identification, expression and crystallization of the protein SjTrx1i and SjTRP14. (A)** 15% SDS-PAGE analysis of SjTrx1i purification by the Ni-NTA affinity chromatography. (M: marker, P: precipitate, S: supernatant, FT: flow-through, W1–4: wash fractions 1–4, E: eluate). **(B)** Elution profile of SjTrx1i from Superdex 200 Increase 10/300 GL size-exclusion chromatography. **(C)** Gel filtration chromatography displaying the purified SjTrx1i protein obtained from both Ni beads affinity chromatography and Superdex 200 size-exclusion filtration. **(D)** Crystals of the SjTrx1i protein. **(E)** 15% SDS-PAGE analysis of SjTRP14 purification by the Ni-NTA affinity chromatography. (M: marker, P: precipitate, S: supernatant, FT: flow-through, W1–4: wash fractions 1–4, E: eluate). **(F)** Elution profile of SjTRP14 from Superdex 200 Increase 10/300 GL size-exclusion chromatography. **(G)** Gel filtration chromatography showing the purified SjTRP14 protein obtained from Ni beads affinity chromatography and Superdex 200 size-exclusion filtration. **(H)** Crystals of the SjTRP14 protein. **(I)** The SjTrx1i gene utilized in this study is an isoform of the previously reported SjTrx1, differing by five amino acid substitutions: K68R, R69K, T73S, R86K, and V101A.
(TIF)

**S10 Fig. Alignment of SjTrx1i and SjTRP14. (A)** Sequence alignment of SjTrx1i and SjTRP14. **(B-C)** The 2*Fo-Fc* electron density maps (1σ, blue meshes) and Fo-Fc electron density maps (3σ, green meshes) for C34 and C37 in the oxidized (PDB: 22FJ) and reduced states (PDB: 22FK) of SjTrx1i are shown. **(D-E)** The 2*Fo-Fc* electron density maps (1σ, blue meshes) and *Fo-Fc* electron density maps (3σ, green meshes) for C41 and C44 in the oxidized (PDB: 22FG) and reduced states (PDB: 22FH) of SjTRP14 are presented.
(TIF)

**S11 Fig. Comparative analysis of the active sites in SjTrx1i and SjTRP14.** The active sites of the SjTrx1i **(A)** and SjTRP14 **(B)** are presented for both their oxidized and reduced states. Distance measurements are indicated by black dashed lines.
(TIF)

**S12 Fig. Analysis of worm burden and body weight in mice. (A-B)** The number of female or male worms recovered at 42 dpi in GFP (RNAi), SjTRP14 (RNAi), and SjTGR (RNAi) groups. **(C-D)** Changes in relative body weight and percentage of weight gain among the three groups. For all the mouse experiments, each group contained six biological replicates, and the data are presented as mean±SD.
(TIF)

**S1 Table. Overview of TGR Structures currently available in the PDB.**
(DOCX)

**S2 Table. List of primers used in this study.**
(DOCX)

**S3 Table. Crystallographic data collection and refinement statistics for SjTGR$^{WT}$ and SjTGR$^{U597C}$.**
(DOCX)

**S4 Table. Crystallographic data collection and refinement statistics for SjTGR-NADPH, SjTGR-GSH and SjTGR-Au.**
(DOCX)

**S5 Table. Anomalous scattering signals of Au.**
(DOCX)

**S6 Table. Crystallographic data collection and refinement statistics for SjTrx1i and SjTRP14.**
(DOCX)

**S7 Table. Quantification of Selenium incorporation efficiency by ICP-MS.**
(DOCX)

**S1 File.**
(DOCX)

## Acknowledgments

We are grateful to the staff of the BL-02U1, BL-10U2, and BL-19U1 beamlines at the National Center for Protein Sciences Shanghai (NCPSS) at Shanghai Synchrotron Radiation Facility (SSRF). We would like to express our gratitude to Prof. Elias Arnér and Dr. Qing Cheng (Karolinska Institutet, Sweden) for providing the pABC2a plasmid and sharing their methodologies for the expression of selenoproteins.

## Author contributions

**Conceptualization:** Songqing Wang.

**Data curation:** Songqing Wang, Shukun Zhong, Yuxuan Huang.

**Formal analysis:** Wenbin Hong, Zhijian Liang, Chuchu Zhang, Xianshu Liu, Ziyi Dai, Siqi Wu, Caiming Wu, Yuxuan Huang, Peicheng Hong.

**Funding acquisition:** Shuaiqin Huang.

**Investigation:** Songqing Wang, Wenbin Hong, Shukun Zhong, Tianyichen Xiao.

**Methodology:** Songqing Wang, Wenbin Hong, Shukun Zhong, Zhijian Liang, Chuchu Zhang.

**Project administration:** Tianwei Lin, Xueqin Chen, Shuaiqin Huang.

**Resources:** Shukun Zhong, Zhijian Liang, Tianyichen Xiao, Chuchu Zhang, Xianshu Liu, Ziyi Dai, Siqi Wu, Peicheng Hong, Haixia Ren, Tianwei Lin, Xueqin Chen, Shuaiqin Huang.

**Software:** Songqing Wang, Tianyichen Xiao, Yunlong Li, Qixu Cai.

**Supervision:** Caiming Wu, Shaowei Li, Tianwei Lin, Xueqin Chen, Shuaiqin Huang.

**Validation:** Wenbin Hong, Zhijian Liang, Ziyi Dai.

**Visualization:** Wenbin Hong, Shukun Zhong, Tianyichen Xiao, Xianshu Liu, Yunlong Li, Qixu Cai.

**Writing – original draft:** Songqing Wang.

**Writing – review & editing:** Tianwei Lin, Xueqin Chen, Shuaiqin Huang.

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
