## [Decision Letter · Decision Letter 0]

23 Dec 2025

PPATHOGENS-D-25-02732

Structural Basis for Substrate Recognition and Inhibition of Thioredoxin Glutathione Reductase from Schistosoma j aponicum : Implications for Antiparasitic Development

PLOS Pathogens

Dear Dr. Huang,

Thank you for submitting your manuscript to PLOS Pathogens. After careful consideration, we feel that it has merit but does not fully meet PLOS Pathogens's publication criteria as it currently stands. Therefore, we invite you to submit a revised version of the manuscript that addresses the points raised during the review process.

We look forward to receiving your revised manuscript.

Kind regards,

Holger Sondermann, Ph.D.

Academic Editor

PLOS Pathogens

Edward Mitre

Section Editor

PLOS Pathogens

Sumita Bhaduri-McIntosh

Editor-in-Chief

PLOS Pathogens

orcid.org/0000-0003-2946-9497

Michael Malim

Editor-in-Chief

PLOS Pathogens

orcid.org/0000-0002-7699-2064

**Additional Editor Comments:**

As you will see, all three reviewers acknowledge the new insight this study provides and its potential impact. However, all reviewers raise important concerns regarding both the experimental work and the presentation, which would need to be addressed in detail.

**Journal Requirements:**

1) Please provide an Author Summary. This should appear in your manuscript between the Abstract (if applicable) and the Introduction, and should be 150-200 words long. The aim should be to make your findings accessible to a wide audience that includes both scientists and non-scientists. Sample summaries can be found on our website under Submission Guidelines:

https://journals.plos.org/plospathogens/s/submission-guidelines#loc-parts-of-a-submission

Potential Copyright Issues:

- Please confirm (a) that you are the photographer of Figure 6., or (b) provide written permission from the photographer to publish the photo(s) under our CC BY 4.0 license.

ii) Figure Figure 6.. Please confirm whether you drew the images / clip-art within the figure panels by hand. If you did not draw the images, please provide (a) a link to the source of the images or icons and their license / terms of use; or (b) written permission from the copyright holder to publish the images or icons under our CC BY 4.0 license. Alternatively, you may replace the images with open source alternatives. See these open source resources you may use to replace images / clip-art:

State what role the funders took in the study. If the funders had no role in your study, please state: "The funders had no role in study design, data collection and analysis, decision to publish, or preparation of the manuscript.".

**Reviewers' Comments:**

Reviewer's Responses to Questions

**Part I - Summary**

Reviewer #1: This study resolves the structures to Schistosoma japonicum thioredoxin glutathione reductase (SjTGR) in combination with various cofactors, substrates, and the anti-schistosomal compound auranofin. The crystal data provide resolution into enzymatic mechanism and new insight into the mechanism of action of auranofin. While structural insight into TGR is already available from other schistosome species (PDB:7B02), the breadth and resolution of this work provides new detail and is a useful comparison for the S. mansoni complex.

Reviewer #2: This manuscript, “Structural Basis for Substrate Recognition and Inhibition of Thioredoxin Glutathione Reductase from Schistosoma japonicum”, provides a comprehensive structural and biochemical analysis of SjTGR, a parasite-specific selenoenzyme essential for redox homeostasis. SjTGR as a drug target for schistosomiasis. The authors present eleven crystal structures of SjTGR in multiple redox states and in complex with NADPH, GSH, and Auranofin, supported by mutagenesis, enzymatic assays, and in vivo RNAi experiments. They show that a unique hook-shaped C-terminal motif stabilized by residue T594 of SjTGR fine-tunes electron transfer. R317 is shown to be critical for NADPH binding, while C28 in the Grx domain is indispensable for GSH-dependent activity. Structural data reveal that the inhibitor Auranofin binds simultaneously to three redox centers, explaining its action. Activity assays of SjTGR’s substrates SjTrx1i and SjTRP14 highlight distinct recognition modes, and RNAi knockdown of SjTGR or SjTRP14 impairs parasite survival, reducing pathology in mice. This is substantial work providing new insight into SjTGR function. However, the analysis of the structural data requires clarification.

Reviewer #3: The authors undertake a thorough structural/functional analysis of potential drug targets for Schistosoma japonicum. The study is overall well presented and designed and for the most part, fairly clearly presented. I congratulate the authors on their interesting work. I do have a variety of minor comments and suggestions below.

**Part II – Major Issues: Key Experiments Required for Acceptance**

Please use this section to detail the key new experiments or modifications of existing experiments that should be absolutely required to validate study conclusions.required to validate study conclusions.

Reviewer #1: Major

While the structural data is well elaborated, a major issue is one of rigor of presentation of the functional data. In many places it is unclear from text and figure legends the number of assays and replicates that have been performed, and as such whether errors are quoted as sd or sem. This needs to be clearly elaborated throughout the manuscript for all substrate assays and especially the animal work, including the in vivo RNAi where methods are poorly described. The numbers of mice and worms (Fig 6B&C), the number of relevant replicates used for qPCR analyses need to be clearly outlined to ensure replicability. Does the sex bias refer to recovered worms or mice (Fig 6B,C)? How many tail injections were performed (10?). If there are 6 mice per group, how many groups are shown? The graph axis in 6D refers to ‘insects’ which is surely incorrect! Are these data normalized to naïve worms or anti-GFP worms?

Reviewer #2: A major advantage of this study is that they employ the native Sec-containing SjTGR for both structural characterization and enzymatic assays. However, it is key to show that indeed the enzyme used in this study was pure and primarily Sec-containing. While the presence of Sec in position U597 is shown in supporting S1, it does not rule out the minor presence of other forms of SjTGR, with possibility of minor misincorporation of serine and other amino acids at position, or truncation at position 597. The purity of the preparation and the percentage of Sec incorporation would influence the kinetic parameters reported in Table 1. An intact protein mass spectrometry of the batches used in the experiments is essential for supporting the conclusions drawn in the manuscript.

The second point is the decision process of deducing the presence or absence of disulfide and selenylsulfide bonds in the redox centers. First the material and methods and table S2 seem to suggest that crystallization was set only under oxidizing conditions without reducing reagents. The text (lines 175-177 states “The structures of SjTGRWT and SjTGRU597C in four redox states were analyzed, which included the fully oxidized state (Eox), two intermediate states reduced with two electrons (EH2), and the fully reduced state with four electrons (EH4).” How were these different redox states EH4, EH2’, EH2 and Eox obtained experimentally? Tables S2, S3 and S5 should either include the PDB IDs and sample conditions or those should be added in a separate panel. The rationale for deciding whether a disulfide or selenylsulfide bond is present or not is not fully explained (distances and electron density?). Mobility could also contribute to missing electron density, so the B factors should be displayed. Sample heterogeneity could also be a factor (a mixture of redox states?). Complicating the ability of the reader to judge the data is the unclear graphics in Figure 1 panels D-G. The electron density should be made more transparent for easier visualization. Avoid the use of yellow for atoms other than sulfur (for example see cofactors) or distances. It would help if the PDB IDs were included in the captions per panel.

All of this makes evaluating the data confusing. While Figure 1 panel I is considered the reduced state of the wild type (C-terminus residues GCUG), to this reviewer it appears to be in the same conformation as the oxidized form of the GCCG variant. Certainly, the CU Figure S3I appears to be in a selenysulfide bond and not reduced. Indeed, the high reactivity of Sec necessitates that the selenylsulfide bond reform rapidly, even if the reduced enzyme was used to set the crystal trays (under conditions reported to be depleted of reducing agents). In 2 days, if crystallization, the selenylsulfide bond CU should be the exclusive form. Since the criteria by which this was decided are not discussed, the reader is left ambiguous whether indeed (lines 186-187) the “C-terminal selenothiol motif (C596-U597/C597) in SjTGR adopted a cis-conformation in both oxidized and reduced states”. It may be because the CU is defined as reduced in the S3 panel I, even though it is possibly oxidized.

The supporting information is incomplete and does not provide enough detail to allow the experiments to be reproduced. The proteins were cloned into pET28a, but the location of the His tag (N- or C-terminal) is not specified and needs to be inferred from the structures. The affinity tag does not appear to have been removed, yet its potential impact on enzymatic activity is not addressed. The SECIS element following the GCUG* is also not described. Was it the native Schistosoma japonicum SECIS or an E. coli formate dehydrogenase SECIS element? The authors note a five–amino acid difference in the SjTrx1i sequence compared to previous reports, but the specific amino acids are not specified. Without a complete description of the amino acid sequences and plasmid constructs, these results could not be reproduced. Overall, a more thorough and detailed Materials and Methods section is required.

Related to that, error bars and significant figures should be consistent. There are multiple figures where it is not clear whether error bars are not shown because they are missing or very small. For example, panels 1L and 42G, Figure 2G (if it is a representative measurement, add the other two to the supporting information), panels 6A-B, and other panels in the supporting information. Was all data acquired in triplicate? There are no error estimations in Table 1. Overall, the manuscript would benefit from clarification when error bars are too small to be displayed. Lastly, Error analysis should report significant figures consistently. In Biochemistry, we typically keep one or two significant figures. For some measurements, like the ITC, it is hard to imagine that the error can be so precisely estimated as to have three significant figures. For example, line 295 “Kd value of 11.2 ± 3.82 μM” should be rounded to 11.2 ± 3.8 μM.

Reviewer #3: (No Response)

**Part III – Minor Issues: Editorial and Data Presentation Modifications**

Reviewer #1: Minor

Figure 4. The authors utilize various TGR inhibitors in DTNB substrate assays, but report only the structure with auranofin. Please expand on why these other inhibitors are relevant, are the authors implying these inhibitors are of too low affinity to crystallize?

Figure 6. Given data showing SjTGR has a higher catalytic rate for reduction of Trxli over TRP14, it is unclear why RNAi assays focused on SjTRP14

Line 59. ‘knowledge’-based design

Line 89. see PMID 34936384

Line 154. reminiscent

Line 155. Also needs a reference from the literature.

Reviewer #2: From the introduction it is not easy to know which TGR structures are already available. Only one? Consider adding a table t supporting. What is already known about its reaction mechanism? Figure 3S suggests that the introduction can be expanded.

In line 204 is 13% change in kcat/km significant enough to call T594 a modulator (see line 206).

Line 273 “However, its structure basis of its function remained elusive.”

There is a structure of auranofin (Au) with TrxR so it is more clear to state that -

However, its structure basis of its interactions with TGR is unknown.

Line 329 about oxytocin conflicts with observations reported in reference 32 (line 322). Please comment on the differences between the activity assays here and in reference 32.

Line 504 a reference describing C321.ΔA is needed.

Line 548-549 “To determine the reduced structures of SjTrx1i and SjTRP14, the crystals were soaked in 5 mM DTT for 10 minutes” and immediately frozen?

For Fig 1L the caption describes “performed using DTNB as the substrate, measuring TNB production at 421 nm” but figure lists 412 nm. Which wavelength was used?

Supporting:

Clarify that Figure S1B-E is exclusively describing SjTGR U597C.

Provide similar information for the wild-type SjTGR.

Add PDB ID to captions

Figure S6, clarify that these are three independent repeats and that error bars are too small to be observed?

Typos:

Unclear sentences:

Sentence 115 consider changing “deciphering the mechanisms of TGR’s catalysis” to in depth studies of TGR’s reaction mechanism.

Line 153 instead of mutant use variant

A change in nucleic acid is a mutant

A change in amino acid is a variant

Also use the term selenylsulfide bond for Se-S

Line 188 introduce abbreviation Rattus norvegicus TrxR as RnTrxR1. Same for BmTrxR1 line 217. Introducing the abbreviation in figure 1 caption is not sufficiently clear.

Line 196 substitute was with is?

Line 201 substitute “was with” with exhibited

Line 209 “The complex structure of SjTGR and NADPH” consider simplifying to The structure of SjTGR complexed to NADPH

Line 322 Although hTrx1 can reduce insulin, hTRP14 cannot, although both can reduce both oxytocin and vasopressin.

Consider more targeted keywords.

Line 336 “These results suggested that SjTrx1i and SjTRP14 were for different substrates.” These results suggested that SjTrx1i and SjTRP14 have different substrate specificity.

Reviewer #3: Fig S1 - Authors need to define the column names on their gel in panel B (what are FT, W1, W2, etc?).

Fig S2 - the use of Figure panel labels "A" and "B" within panel 2A is this figure is a little confusing. Suggest using roman numerals for the subpanels or just referring to them as left vs right in the figure legend. The author also haven't used this format consistently among the panels within this figure or between figures. I suspect this is included in error, but please check.

Line 173-174 "which was crucial for maintaining TGR functionality in parasitic flatworms." - is this statement relating to the results of this current study or prior literature? If the former, please explain how this was determine. If the latter, please provide citations to support the statement.

Lines 196-197 - the authors reference a glycine residue in other platyhelminths in Fig 1K, but this data is not shown in the alignment. Is this informatin from the literature? Please include it.

Lines 200-202 - The wording is a little confusing here. Are the authors stating that their data showed the dfference in DNTB reducatase activity between the wild-type and mutated form or that the structure analyses predict that this should be the case? I presume the former given the figure panels, but the statement 'would abolish' is a bit ambiguous.

Fig 2C and elsewhere - the use of yellow lettering over the white background of the figure is quite difficult to see. Maybe a darker yellow shade or another darker colour would be better?

Lines 271-273 - I presume this statement is related to prior literature. Please cite the relevant studies presuming that is the case.

Fig 4 - As above, the user of 'A' and 'B' sublabels within some figure panels but not others and the similar use of 'A' and 'B' for different panels within the figure as a whole is a bit confusing. I suggest either denoting left vs right in the figure legend or using another sublabel, but then using it consistently. For example 'A' and 'B' are used to differentiate the left vs right figures in panels 4A, B, C, but not 4D.

Fig S7- As above the column lables in Fig S7A and E need to be defined.

Fig 5 - what do the authors make of the observation that mutating T594, which the authors note (lines 197-198) provides a unique contribution to the conformation of SjTGR and the t594V mutation altered the catalytic effiency of SjTGR (lines 204/205), doesn't affect insulin reduction? Is this expected? The authors show the binding data for T594V in figure 5B, but do not discuss it.

Fig 5J the black text of the magenta protein skeleton is quite difficult to read. Maybe white text where there is an overlap, or arrange the labels outside the protein model with dash lines?

Overall comment - the paper would benefit from a little more context around so of the data included in some figure panels vs data left out or maybe not gathered. For example, there may be a perfectly reasonable explanation for why the authors present the binding affintity of Sj001 to SjTGR in Figure 4H but not similar binding affinity data for Auranofin or TRi-1, but they don't provide any context for this decision. Similarly, are the mutants included in Fig 5B different from the ones included in 5C. Both panels look at the capactiy fo SjTGR mutants to reduce SjTrk1i, but the latter examined behaviour under DTT conditions. Again, there may be an clearl explanation, but the authors don't appear to provide it. In another example. why were C41S and C44S mutants included in 5H? Both, why these mutations in 5H, and why not these mutations in 5E-G? Again, there may likely be a clear reason for this, but some context needs to be provided.

Lines 365-383 - are the phenotypic outcomes identified through SjTGR or TRP14 silencing similar to the effects seen with the application of auranofin or other inhbitors? Has this been examined in previous studies?

Fig 6d - what is meant by 'adult insects' in the figure legend for 6D. I assume this is a typo.

Fig 6 - What are the error bars in the various bar charts in this figure? Standard Deviation, Standard Error? Or something else? Please specify in the figure legend.

PLOS authors have the option to publish the peer review history of their article (what does this mean?). If published, this will include your full peer review and any attached files.). If published, this will include your full peer review and any attached files.

.

Reviewer #1: No

Reviewer #2: No

Reviewer #3: No

**Figure resubmission:**
---

## [Decision Letter · Decision Letter 1]

26 Mar 2026

Dear Prof. Huang,

We are pleased to inform you that your manuscript 'Structural Basis for Substrate Recognition and Inhibition of Thioredoxin Glutathione Reductase from Schistosoma japonicum : Implications for Antiparasitic Development' has been provisionally accepted for publication in PLOS Pathogens.

Best regards,

Holger Sondermann, Ph.D.

Academic Editor

PLOS Pathogens

Edward Mitre

Section Editor

PLOS Pathogens

Sumita Bhaduri-McIntosh

Editor-in-Chief

PLOS Pathogens

orcid.org/0000-0003-2946-9497

Michael Malim

Editor-in-Chief

PLOS Pathogens

orcid.org/0000-0002-7699-2064

Reviewer Comments (if any, and for reference):

Reviewer's Responses to Questions

**Part I - Summary**

Reviewer #2: (No Response)

Reviewer #3: I thank that authors for their thorough response to my comments in revision. I am satisfied with the changes and have no further comments on the manuscript. My congratulations to the authors on their study.

**Part II – Major Issues: Key Experiments Required for Acceptance**

Please use this section to detail the key new experiments or modifications of existing experiments that should be absolutely required to validate study conclusions.required to validate study conclusions.

Reviewer #2: (No Response)

Reviewer #3: (No Response)

**Part III – Minor Issues: Editorial and Data Presentation Modifications**

Reviewer #2: (No Response)

Reviewer #3: (No Response)

PLOS authors have the option to publish the peer review history of their article (what does this mean?). If published, this will include your full peer review and any attached files.). If published, this will include your full peer review and any attached files.

.

Reviewer #2: No

Reviewer #3: No

---

## [Editor Report · Acceptance letter]

Dear Prof. Huang,

We are delighted to inform you that your manuscript, "Structural Basis for Substrate Recognition and Inhibition of Thioredoxin Glutathione Reductase from Schistosoma japonicum : Implications for Antiparasitic Development," has been formally accepted for publication in PLOS Pathogens.

Best regards,

Sumita Bhaduri-McIntosh

Editor-in-Chief

PLOS Pathogens

orcid.org/0000-0003-2946-9497

Michael Malim

Editor-in-Chief

PLOS Pathogens

orcid.org/0000-0002-7699-2064